# Monitoring of the reconstruction process in a high mountainous area affected by a major earthquake and subsequent hazards

Chenxiao Tang[1,2], Xinlei Liu[3], Yinghua Cai[4], Cees Van Westen[2], Yu Yang[3,5], Hai Tang[3], Chengzhang Yang[3], Chuan Tang[3]

[1] Institute of Mountain Hazard and Environment, Chinese Academy of Sciences, China

[2] Faculty of Geo-Information Sciences and Earth Observation (ITC), University of Twente, the Netherlands

[3] State Key Laboratory of Geo-Hazard Prevention and Geo-environment Protection (SKLGP), Chengdu University of Technology, China

[4] Sichuan Institute of Land and Space Ecological Restoration and Geological Hazard Prevention, China

[5] Station of Geo-Environment Monitoring of Chengdu, China.

*Corresponding to*: Chenxiao Tang (c.tang@imde.ac.cn)

**Abstract.** Recovering from major earthquakes is a challenge, especially in mountainous environments where post-earthquake hazards may cause substantial impacts for prolonged periods of time. Although such phenomenon was reported in the 1923 Kanto earthquake and the 1999 Chi-chi earthquake, careless reconstruction in hazard-prone areas and consequently huge losses was witnessed following the 2008 Wenchuan earthquake in Sichuan province of China, as several reconstructed settlements were severely damaged by mass movements and floods. In order to summarize experiences and identify problems in the reconstruction planning, a monitoring of one of the settlements, Longchi town, was carried out by image interpretation and field investigation. Seven inventories containing buildings, farmlands, roads and mitigation measures were made to study the dynamics in element-at-risk and exposure over a period of 11 years. It was found that the total economic value of the new buildings in was several times more than the pre-earthquake situation in 2007, because of enormous governmental investment. Post-seismic hazards were not sufficiently taken into consideration in the recovery planning before the catastrophic

debris flow disaster in 2010. As a result, the direct economic loss from post-seismic disasters was slightly more than the loss caused by the Wenchuan earthquake itself. The society showed an impact - adapt pattern, taking losses from disasters and then gaining resistance by abandoning buildings in hazard-prone areas and installing mitigation measures. The areas potentially expose to post-earthquake hazards were summarized and a possible time table for reconstruction was proposed. Problems might be encountered in hazard assessment and possible solutions were discussed.

Keywords: Earthquake; Reconstruction; element-at-risk; exposure; Wenchuan earthquake;

# 1 Introduction

## 1.1 Background

Major disasters, such as earthquakes, have large impacts on societies, causing massive direct and indirect losses. Large earthquakes may also seriously affect the natural environment, in the form of secondary hazards. In mountainous regions one of the most severe secondary hazards is the triggering of co-seismic landslides. These may result in the loss of vegetation and the production of large volumes of landslide deposits, which drastically change the susceptibility to rainfall-induced mass movements and flooding after the earthquake (Fan et al., 2019a;Fan et al., 2019b;Tang et al., 2016;Yang et al., 2018;Guo et al., 2016). An amplifications followed by a gradual decay in hazards were witnessed after the 1923 Kanto earthquake in Japan (Koi et al., 2008;Nakamura et al., 2000), the 1993 Finisterre earthquake in Papua New Guinea (Marc et al., 2015;Stevens et al., 1998), the 1999 Chi-Chi earthquake in Chinese Taipei (Lin et al., 2006;Shieh et al., 2009;Shou et al., 2011;Chen and Hawkins, 2009), and the 2008 Wenchuan earthquake in PR China (Fan et al., 2018;Fan et al., 2019a;Tang et al., 2019;Tang et al., 2016). The process could last from 6 (Hovius et al., 2011) to about 40 (Nakamura et al., 2000) years.

In addition to the prolonged effect, different post-seismic hazard types may interact with each other, forming hazard chains and further adding complexity to the situation. The most commonly witnessed cases includes landslides forming barrier lakes which later cause outburst floods (Fan et al., 2012;Dong et al., 2011) or debris flows (Hu and Huang, 2017). Moreover the debris flows could result in river

damming and river bed rise as well (Ni et al., 2014;Tang et al., 2012;Xu et al., 2012;Fan et al., 2019b), causing floods. A comprehensive summary of post-earthquake hazard chains was made by Fan et al. (2019b).

Rebuilding and recovering social functions in such circumstances are difficult tasks, as settlements face continuous threats of landslides, debris flows and flash floods. Based on how risk is calculated (van Westen et al., 2006;Fell, 1993;Varnes, 1984), the amplification in hazards and reconstruction would bring sharp changes in risk. Careless planning could result in a large increase in risk and consequently taking severe losses. It has been reported that post-earthquake hazards caused severe damages in Chinese

Taipei after the 1999 Chi-Chi earthquake (Lin et al., 2004;Cheng et al., 2005) and in Sichuan province of PR China after the 2008 Wenchuan earthquake (Tang et al., 2012;Xu et al., 2012;Zhang and Zhang, 2016), but there is a lack of studies summarizing the experiences and problems encountered during the relief and reconstruction periods. It is also not clearly stated when and where to rebuild in such mountainous regions.

To fill this knowledge gap, we conducted a study concerning the recovery in an area hit by the 2008 Wenchuan. Seven inventories of elements-at-risk from satellite images covering a period of 11 years (2007 - 2018) were generated to study the dynamics in exposure and recovery process. The aim is to show encountered problems during the recovering process and propose possible solutions, in order to provide knowledge for future reconstruction efforts in earthquake-susceptible regions.

**1.2 The Wenchuan earthquake**

The $M_w$ 7.9 Wenchuan earthquake occurred on 12 May 2008 in Sichuan province, affecting an area of 110,000 $km^2$, most of which consisting of steep mountains with deeply incised valleys. The earthquake triggered a large number of landslides, and estimations varied between 48,000 and 200,000 (Tanyas et al., 2019;Xu et al., 2014;Dai et al., 2011). Around one third of the 87,537 casualties was estimated to

have been caused by the landslides and not by ground shaking only (Wang et al., 2009a). The estimated losses from the earthquake were around 115 billion US dollar (Dai et al., 2011). After the relief stage the reconstruction began in 2009, and 19 of the Chinese provinces supported each one of the affected counties or cities in the recovery by using at least 1% of their annual provincial revenue for a period of 3

years (Huang et al., 2011;United Nations Office for Disaster Risk Reduction (UNISDR), 2010;Dunford

and Li, 2011;Zuo et al., 2013). The provinces were requested to provide specialists in planning and

design, as well as construction workers. A fast reconstruction progress was witnessed and the

reconstruction was completed in 2012.

Extreme rainfall events in the years following the earthquake triggered numerous mass movements,

mostly in the form of debris flows, destroying many of the reconstructed buildings. One of the most

devastating events occurred in Qingping village (Mianzhu County) on 13 August 2010, when two debris

flows from the Wenjia watershed, destroyed the mitigation measures and buried most of the valley,

including newly reconstructed villages and roads (Tang et al., 2012). Another example of a major

post-earthquake disaster was the debris flow that dammed the Minjiang River which flooded the nearby

Yingxiu town on 14 August 2010 (Xu et al., 2012).   A third major disaster occurred on 10 July 2013,

when a debris flow formed by a breached landslide dam severely damaged the reconstructed buildings in

Qipangou village, destroying most of the farmlands (Hu and Huang, 2017). The losses caused by these

disasters have resulted from a lack of experience in post-earthquake reconstruction planning.

The catastrophic debris flows were caused by the entrainment of co-seismic mass wasting by surface

runoff. In the epicentral area of the 2008 Wenchuan earthquake, the mass movement activities were

highly active in the first three years, and then decayed rapidly (Tang et al., 2016;Yang et al., 2017;Yang

et al., 2018;Zhang et al., 2016). Similar recovery patterns were also observed in the other regions (Li et

al., 2016). The decay is not a linear progress as it is largely affected by the precipitation (Tang et al.,

2016;Fan et al., 2019b). On Aug 20 2019, debris flows again caused severe damages in the Wenchuan

area, suggesting the mass movements were still enhanced.

The Wenchuan earthquake has inspired many studies related to assessing vulnerability and losses (Wang

et al., 2009b;Wu et al., 2012), such as physical (Cui et al., 2013),   social (Hu et al., 2010;Kun et al.,

2009;Lo and Cheung, 2015;Wang et al., 2015;Yang et al., 2015), environmental (Yang et al., 2017),

institutional (Hu et al., 2010), and economic vulnerability (Wu et al., 2012;Zhang et al., 2013).

Household vulnerability was studied in particular by a number of studies (Sun et al., 2010a;Zhang,

2016;Sun et al., 2010b) which included subjective perceptions (Yang et al., 2015), factor analysis on

household vulnerability (Wang et al., 2015) and on household income (Sun et al., 2010b), and household vulnerability to poverty (Sun et al., 2010a). Recovery was studied by (Dalen et al., 2012) and (Wang et al., 2015). But little has been investigated on how effective was the reconstruction and how much property value was exposed to the post-earthquake hazards due to careless planning.

**1.3 Study area**

The study was conducted in the Longxi watershed, located within 20 km from the epicenter of the 2008 Wenchuan earthquake in Sichuan province of China (Fig. 1). The valley had 2306 permanent residents based on the national census in 2010 (Baidu Encyclopedia, 2016). The area of the watershed is about 89 $km^2$ and the elevation ranges from 810 to 3200 m. The main channel of the Longxi River, which is a

tributary of the Minjiang River, has an average yearly discharge of 3.44 $m^3$/s and the recorded maximum discharge was 300 $m^3$/s. The river flows through the Zipingpu hydropower reservoir which is also one of the major water sources of the province, providing drinking water to the large city of Chengdu (with 16.3 million inhabitants). The climate is sub-tropical, with an average annual precipitation of 1135 mm, of which 80% occurs from May to September. The highest precipitation takes place in August with a

maximum recorded intensity of 83.9 mm/h (Sichuan Geology Engineering Reconnaissance Institute, 2010).

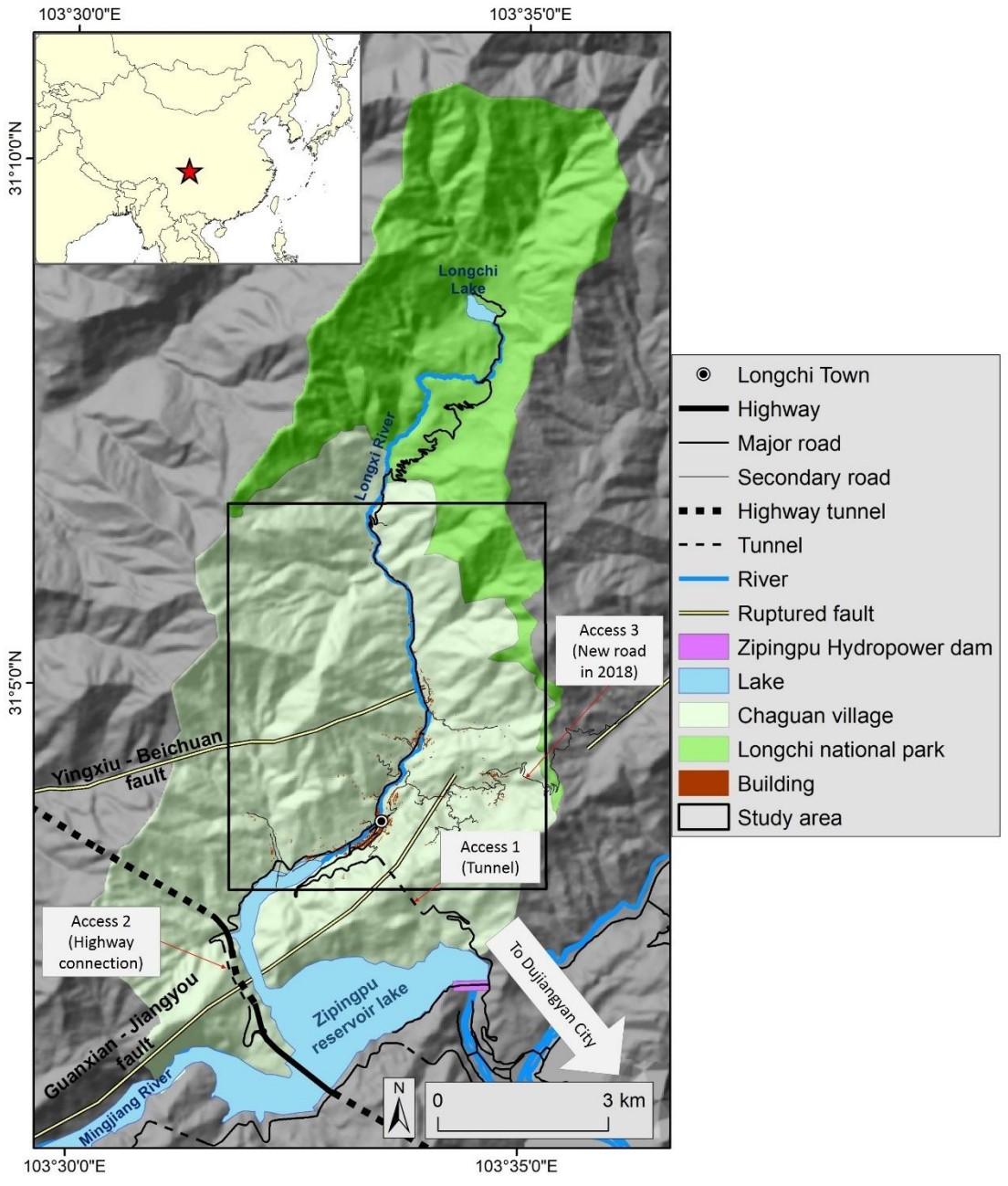

**Figure 1: The location of the study area, Longxi watershed, which contains most of the buildings in the watershed. The roads and buildings reflect the situation in 2018. Buildings outside the study area polygon were not mapped.**

One of the two major faults that ruptured during the earthquake passes through the area: the Yingxiu – Beichuan fault, which had a horizontal displacement of 4.5 m and a vertical displacement of 6.2 m (Gorum et al., 2011). The Guanxian – Jiangyou fault in the south was ruptured during the earthquake as

well (Li et al., 2010). As shown in Fig. 1 the surface ruptures splits into two branches in this region. At three kilometers the surface rupture continues in the eastern side of the watershed. Most of the area is underlain by granite, with some conglomerate distributed in the north, and carbonatite and sandstone in the south.

## 2 Data & methodology

In order to monitor the changes in the post-earthquake period, we acquired a series of ten high (5 -10 m) to very high (0.5 - 2.5 m) resolution satellite images covering the period between 2005 and 2018 (Table 1).

| Data type | Data source | Collection date | Cell size Pan/Mul (m) | Band |
|---|---|---|---|---|
| Satellite images | Quickbird | JUL 2005 | 2.4 | Mul |
| | IKONOS | SEP 2007 | 1 | RGB |
| | Aerial photographs | JUN 2008 | 1 | RGB |
| | Spot 5 | FEB 2009 | 2.5/10 | Pan + Mul |
| | Worldview-2 | MAR 2010 | 0.5/2 | Pan + Mul |
| | Worldview-2 | APR 2011 | 0.5/2 | Pan + Mul |
| | Pleiades | APR 2013 | 0.5/2 | Pan + Mul |
| | Pleiades | DEC 2014 | 0.5/2 | Pan + Mul |
| | Spot 6 | APR 2015 | 1.5 | RGB |
| | Pleiades | JUN 2018 | 0.5/2 | Pan + Mul |
| DTM | Aerial LiDAR | 1999 | 5 | - |
| Landslide inventory | Tang et al. (2016) | 2016 | Polygon-based vector data with landslide activity mapped for 5 periods (2008 - 2015) | |
| | This study | 2018 | Polygon-based inventory based on image from June 2018 | |

**Table 1. Data used for interpretation (Pan= panchromatic image, Mul = multi-spectral image, RGB = Red/Green/Blue: color composite).**

The images were georeferenced with Erdas IMAGINE Autosync Workstation and ARCMAP Geo-referencing Tool. A LiDAR DTM provided by the National Bureau of Surveying and Mapping of China was used to visualize images in a 3D environment in ArcScene software to assist interpretation. The multi-temporal landslide inventories reported in Tang et al. (2016) were used to identify the active landslides over time. An additional landslide inventory was made for 2018, to match with the mapping of the buildings, roads and landside in this study using the Pleiades image from June 2018.

Before interpreting built-up areas, we also consulted OpenStreetMap, in order to evaluate if data from this platform could be used. Unfortunately, the information in OpenStreetMap was very general for the Wenchuan earthquake-affected area, and was limited to the main roads, and general polygons of settlements. Given the current difficulty to digitize and store data in OpenStreetMap from different time periods we decided to generate our database outside of the platform.

We used the above mentioned data to interpret and digitized manmade features, including buildings, farmlands, plantations, roads and mitigation works. Inventories were made for the following years: 2007, 2008, 2010, 2011, 2013, 2015 and 2018. The inventory of 2007 was made first, then the 2008 inventory was created based on modifying the earlier inventory using the aerial photograph of 2008. The inventory of 2010 was derived by modifying the inventory of 2008 using the Wolrdview-2 image from 2010, and the inventory of 2011 was derived from the 2010 inventory, and so on. Digitizing in such a manner allowed us to keep consistency among the multi-temporal inventories. A series of attributes listed in Table 2 were acquired for the digitized features through image interpretation, field mapping, and interviews.

With the help from the Station of Geo-environment Monitoring of Chengdu, we were able to interview the local authorities about historical events and access some of their documents regarding rural planning and population. combining their descriptions and records with our field investigation, buildings in this region were classified based on their functions, construction types, and builders.

*Residences* are buildings to accommodate locals or workers attending the relief and the reconstruction. *Hostels* were to provide accommodation and recreation for tourists. *Institutional buildings* refer to public service buildings like schools, hospitals and water pumping stations. *Commercial buildings*

accommodate shops and local companies. *Agricultural buildings* are used for storage of livestock, agricultural products and farming equipment. *Shelters* are temporary residences, including pre-fabricated

houses, tents and shacks.

A total of seven building construction types were found in this region, including three types served as temporary shelters. *Reinforced concrete frame* (RCF) (Fig. 2 A), *reinforced masonry* (RCM) (Fig. 2 B), *wood and brick* (WB) (Fig. 2 C), *wooden* (W) structures (Fig. 2 D1 & D2) are permanent buildings, and *pre-fabricated metal houses* (PFM) (Fig. 2 E), tents (Fig. 2 F1), and shacks (Fig. 2 F2) served as

temporary shelters. *Tents and shacks* were categorized as one type in this study due to similar cost and size.

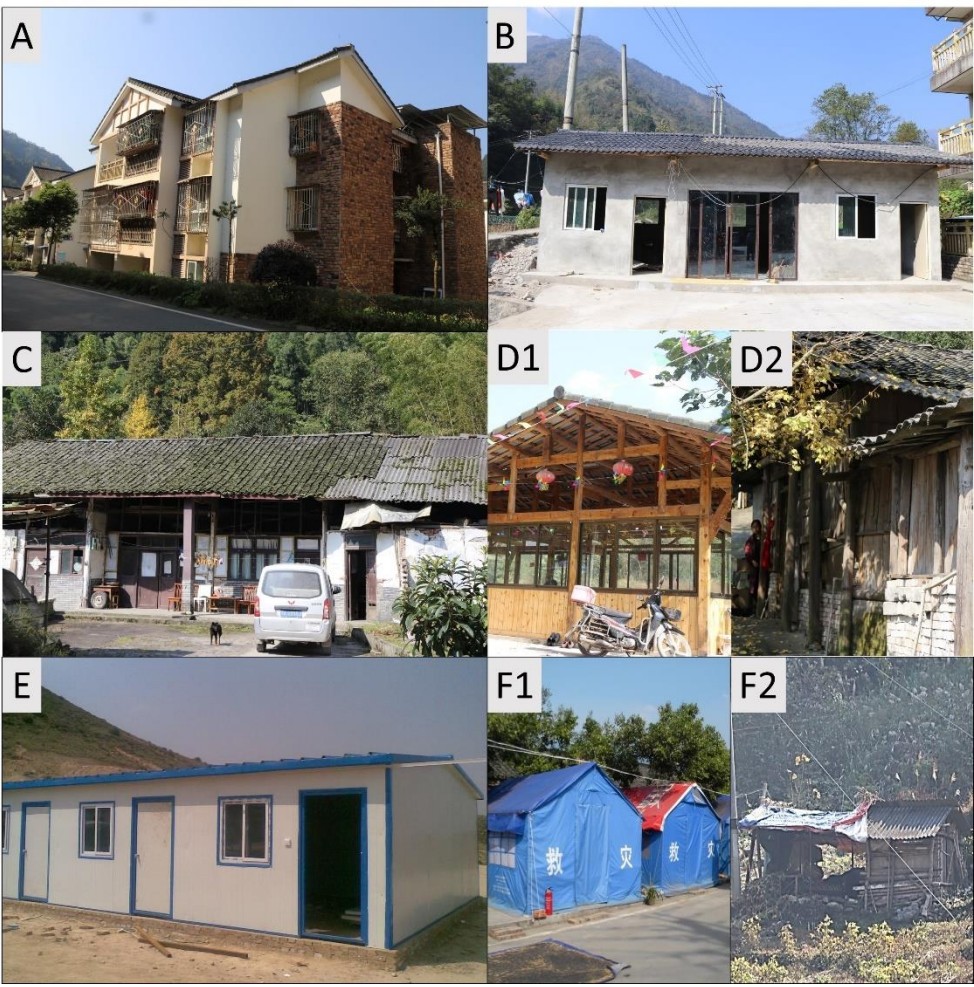

**Figure 2:  Examples of building construction types in the study area.  A: RC frame (RCF) structure residences built by the reconstruction teams from Shanghai city. B: reinforced masonry (RCM) building of a**

**hostel. C: wood and brick residence (WB). D1: wooden structure (W) serving as restaurant. D2: wooden residence with walls made by wooden plates and bricks. E: pre-fabricated metal (PFM) temporary houses. F1: tents distributed by the government. F2: a shack made from wood, asbestos tiles and waterproof cloth.**

Farmlands were classified into *crops for food* or *commercial crops*. Commercial crops are several local plant species, including kiwifruit, tea, and *magnolia officinalis*, that were widely cultivated and exported to benefit the local economy. Crops for food are the vegetables grown for local consumption.

Roads were categorized into: *major road*, which were wide and built by the national government; *secondary road*, which is narrower than the major road and could be either local-build or constructed with help from the government; *dirt roads* are roads without asphalt or concrete layer. Several *bridges* and *tunnels* were mapped as well.

Mitigation works were mapped, and were classified into: *check dams*, which block debris flow runout and slow down erosion; *drainage channels* are used to redirect runout of debris flows and floods into river directly, avoiding flow through built up areas; *embankments* are built to shield of debris flow and flood runout; *reinforced slopes* are stabilized with reinforcement measures and sometimes combined with drainages.

The status of a building is determined by the attributes of damage level, damage type and usage status. The *damage level* indicates the magnitude of damage a building receives and was assigned based on both image observation and interviewing local people and authorities. If a building is not damaged, *level 0* is assigned. Moderately damaged (*level 1*) means a disaster-affected building was damaged and restored its function after repair. If a building was damaged beyond repair and not collapsed, it was considered as *severely damaged (level 2)*. If a building collapsed, it was classified as destroyed (*Level 3*).

The *damage type* shows what type of hazard feature affected the building, such as ground shaking, landslide, debris flow and flooding. Under certain circumstances a building could be affected by more than one hazard type, for instance by ground shaking and landslide impact at the same time. The *usage status* indicates if a feature is functioning normally, is temporary not been used, or completely abandoned. It is assigned based on field mapping and interviews. The *geometrical* attributes (area or length) were calculated automatically in ArcMap, based on the polygon (buildings or land parcels) or line

(road) features. *Floor space* was calculated by multiplying the number of building floors with the footprint area. The *unit price* is the cost to construct buildings per square meter and was obtained through interviews, and literature study. The *replacement value* of a building was estimated by multiplying the unit price with the floor space. All the economic values in this study were converted to US dollar (USD) with a 10-years-average exchange rate of 1 dollar = 6.51 Chinese Yuan.

We investigated economic recovery by interviewing the local inhabitants and village authorities. Unfortunately, most of them were not willing to share information regarding their income, thus we could only make a descriptive analysis. Each of the interviewees represents one family in the analysis. A total of 113 persons were interviewed in 2018.

| Attributes | Varieties / descriptions | Source | | | | |
|---|---|---|---|---|---|---|
| | | Image | Mapping | Interview | Literature | Calculated |
| Buildings | | | | | | |
| Construction types | **Permanent buildings**: RC frame structure / Reinforced masonry / Wood & brick / Wooden **Temporary buildings**: Pre-fabricated metal houses / Tents & shacks | x | x | x | | |
| Function | Residence / Hostel / Institutional / Commercial / Agricultural building / Shelter | | x | x | | |
| Builder | Self-constructed / government-build | | x | x | | |
| Unit price | 150 – 2700 Chinese yuan per $m^2$, depending on Construction types | | | x | x | |
| Building floors | Floors of a building. A maximum of 4 floors was allowed. | | x | | | |
| Floor space | Building area * building floors | | | | | x |
| Value | Floor space * unit price | | | | | x |
| Roads | | | | | | |
| Type | Major road / Secondary road / Dirt road / tunnel | x | x | | | |
| Farmlands | | | | | | |
| Type | Food crops / Commercial crops | x | x | | | |
| Mitigation works | | | | | | |
| Type | Check dam / Drainage channel / Embankment / Reinforced slopes | x | x | | | |
| all elements-at-risk | | | | | | |

| | | | | | | |
|---|---|---|---|---|---|---|
| Damage level | No damage / Moderately damaged / severely damaged / Destroyed | x | x | | | |
| Damage type | Earthquake / Slides / Debris flows / Flood / No damage | x | x | x | | |
| Usage status | Normal / Abandoned / Empty | x | x | | | |
| Geometry | Auto calculated in ArcMap | | | | | x |

**Table 2. Attributes of the element-at-risk inventories, and the main methods of collection (Image = Image interpretation, Mapping = field mapping, Interview= Interviews with local people and authorities, Literature = various published and unpublished sources, Calculated = calculated from other attributes**

## 3 Monitoring reconstruction

In this section we monitor the changes of the built-up environment caused by human activities and disasters from 2007 to 2018. The overall statistics are shown in Table 3.

| Period | Land use | Construction type | | | | | | |
|---|---|---|---|---|---|---|---|---|
| | | RCF | RCM | WB | W | PFM | TSs | Total |
| 2007: pre-earthquake | Residences | 0 | 66(12) | 186 | 51 | 0 | 0 | 304(12) |
| | Hotels | 1 | 75 | 10 | 0 | 1 | 0 | 87 |
| | Institutional building | 3(3) | 1(1) | 0 | 0 | 0 | 0 | 4(4) |
| | Agricultural | 0 | 0 | 0 | 23 | 0 | 0 | 23 |
| | total | 4(3) | 142(13) | 196 | 74 | 1 | 0 | *417(16) |
| 2008: shortly after the earthquake | Residences | 0 | 24 | 40 | 5 | 0 | 0 | 69 |
| | Hotels | 0 | 9 | 0 | 0 | 0 | 0 | 9 |
| | Institutional building | 2(2) | 0 | 0 | 0 | 0 | 0 | 2(2) |
| | Agricultural | 0 | 0 | 0 | 1 | 0 | 0 | 1 |
| | Shelters | 0 | 0 | 0 | 0 | 82(82) | 227 | 309(82) |
| | Total | 2(2) | 33 | 40 | 6 | 82(82) | 227 | *390(84) |
| 2010: earthquake reconstruction almost | Residences | 126(118) | 78 | 237 | 42 | 0 | 0 | 483(118) |
| | Hotels | 77 | 18 | 2 | 3 | 0 | 0 | 100 |
| | Institutional building | 25(25) | 1(1) | 0 | 0 | 0 | 0 | 26(26) |
| | Agricultural | 0 | 1 | 1 | 86 | 0 | 0 | 88 |
| | Commercial | 36(32) | 2 | 0 | 1 | 0 | 0 | 39(32) |
| | Shelters | 0 | 0 | 0 | 0 | 116(11 | 21 | 137(116) |
| | Total | 266(175) | 99(1) | 239 | 132 | 116(11 | 21 | *873(292) |
| 2011: after devastating debris flows | Residences | 124(116) | 65 | 236 | 40 | 2 | 0 | 467(116) |
| | Hotels | 59 | 12 | 1 | 3 | 0 | 0 | 75 |
| | Institutional building | 25(25) | 1(1) | 0 | 0 | 0 | 0 | 26(26) |
| | Agricultural | 0 | 1 | 7 | 86 | 0 | 0 | 94 |
| | Commercial | 36(32) | 2 | 0 | 1 | 0 | 0 | 39(32) |
| | Shelters | 0 | 0 | 0 | 0 | 50(50) | 3 | 53(50) |
| | Total | 244(173) | 76(1) | 229 | 127 | 52(50) | 3 | *712(224) |
| 2013 all reconstru | Residences | 143(132) | 56 | 206 | 42 | 3 | 0 | 450(132) |
| | Hotels | 68 | 17 | 1 | 3 | 0 | 0 | 89 |
| | Institutional building | 20(20) | 1(1) | 0 | 0 | 0 | 0 | 21(21) |

| | | | | | | | | |
|---|---|---|---|---|---|---|---|---|
| | Agricultural | 0 | 1 | 2 | 76 | 0 | 0 | 79 |
| | Commercial | 36(32) | 2 | 0 | 1 | 0 | 0 | 39(32) |
| | Total | 267(184) | 77(1) | 209 | 122 | 3 | 0 | *678(185) |
| 2015 | Residences | 142(132) | 68 | 199 | 45 | 3 | 0 | 457(132) |
| | Hotels | 69 | 13 | 1 | 3 | 0 | 0 | 86 |
| | Institutional building | 19(19) | 1(1) | 0 | 0 | 0 | 0 | 20(20) |
| | Agricultural | 0 | 1 | 6 | 78 | 0 | 0 | 85 |
| | Commercial | 36(32) | 2 | 6 | 1 | 0 | 0 | 45(32) |
| | Total | 272(183) | 85(1) | 208 | 127 | 3 | 0 | *693(184) |
| 2018 | Residences | 142(132) | 68 | 199 | 49 | 3 | 0 | 461(132) |
| | Hotels | 71 | 13 | 1 | 3 | 0 | 0 | 88 |
| | Institutional building | 19(19) | 2(2) | 0 | 0 | 0 | 0 | 21(21) |
| | Agricultural | 0 | 1 | 8 | 77 | 0 | 0 | 86 |
| | Commercial | 36(32) | 2 | 4 | 1 | 0 | 0 | 43(32) |
| | Total | 268(183) | 86(2) | 212 | 130 | 3 | 0 | *699(185) |

**Table 3:Number of functioning buildings per construction type and land use for the seven time periods considered. The numbers before the brackets indicate the total number and the numbers in the brackets indicate the building numbers built by government. *Sum of all buildings.**

**3.1 The impact of the earthquake (2007 - 2008)**

A total of 417 buildings in 2007 were identified from visual interpretation (Table 3 and Fig. A1) and most of them were self-build residences. Most buildings were not properly designed to withstand a major earthquake (Table 3). The last major earthquake in this area dates back from 1933 (Deixi earthquake), and there were no eyewitnesses alive of that event anymore in 2007. According to investigation reports, debris flows had not been witnessed in 50 years until the Wenchuan earthquake (Yi et al., 2009;Luo et al., 2010;Sichuan Geology Engineering Reconnaissance Institute, 2010, 2011;Sichuan Geological Survey institute, 2010).

The 2008 May 12[th] Wenchuan earthquake triggered 1597 landslides in the study area according to the landslide inventory of Tang et al., (2016). Only a few casualties were reported in this region, as the earthquake occurred at 14:28 when most of the inhabitants were working outdoors.

The earthquake affected 444 buildings including some newly built ones in 2008 (Fig. 3). A total of 142 buildings being completed destroyed (Damage level 3), in which 29 were destroyed by co-seismic landslides. Based on the 2009 SPOT image and the 2010 Worldview-2 images a total of 221 buildings were severely damaged and subsequently removed. The remaining 81 buildings were repaired and

240 functioned normally in 2009 and 2010, thus were classified as moderately damaged. A summary of the

building damage is shown in Table 4.

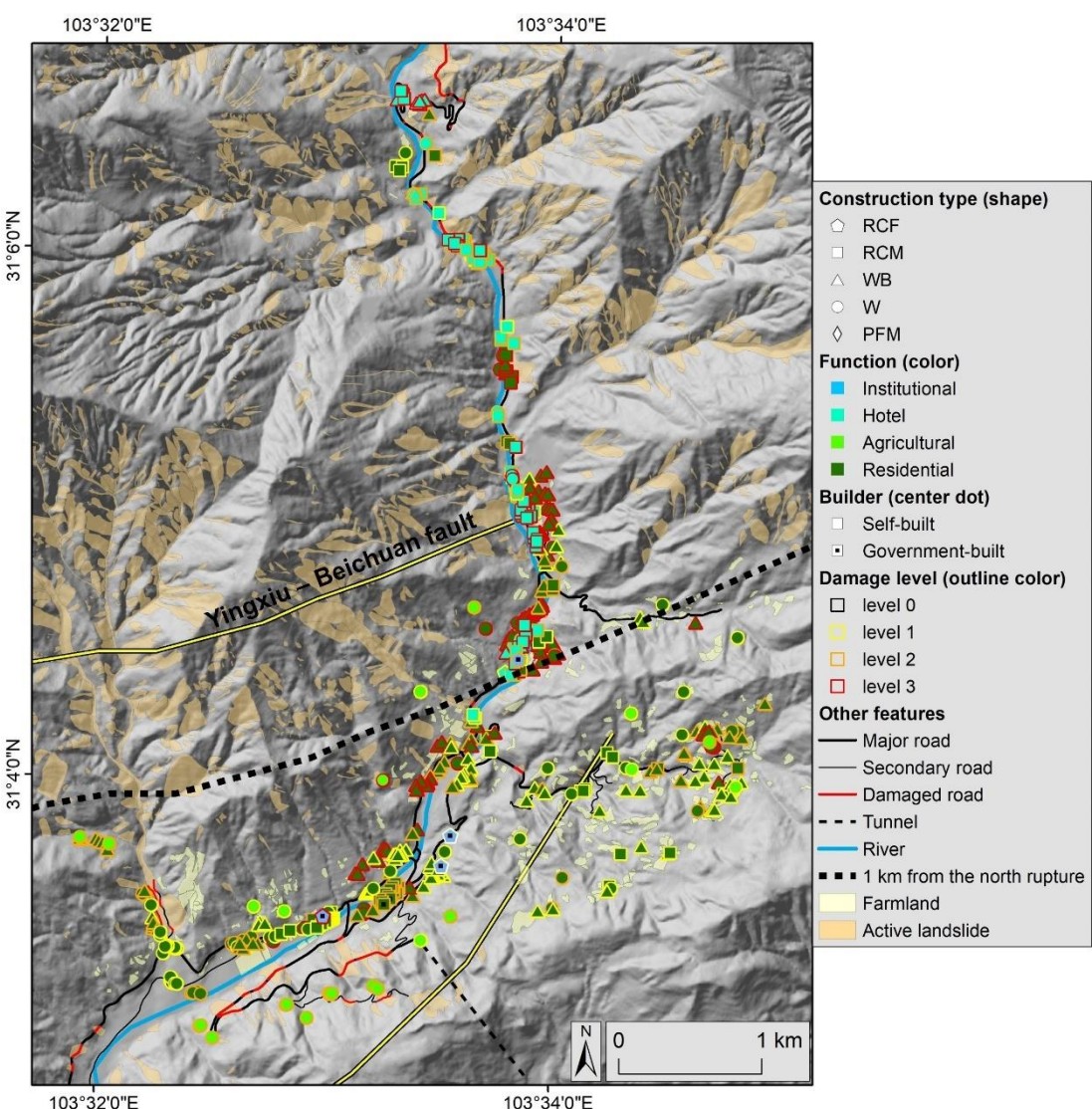

**Figure 3: A map showing the damage level of buildings and the distribution of co-seismic landslides after the 2008 Wenchuan earthquake. Buildings on the foot wall (Southeast) and 1 km away from the north fault**

**rupture (indicated by the thick dotted line) took significantly less damage.**

| Construction type | Floors | Damage levels | | | Sum by floors and construction type |
|---|---|---|---|---|---|
| | | Level 1 | Level 2 | Level 3 | |
| RCF | 1 floor | 2 | 0 | 0 | 2 |
| | 2 floors | 0 | 2 | 0 | 2 |
| RCM | 1 floor | 22(35%) | 19(30%) | 22(35%) | 63 |
| | 2 floors | 11(13%) | 51(63%) | 20(24%) | 82 |
| WB | 1 floor | 34(21%) | 70(44%) | 56(35%) | 160 |
| | 2 floors | 6(9%) | 38(55%) | 25(36%) | 69 |
| W | 1 floor | 6(10%) | 38(60%) | 19(30%) | 61 |
| | 2 floors | 0 | 3 | 0 | 3 |
| Sum by building floors | 1 floor | 64 (22%) | 127(44%) | 97(34%) | 288 |
| | 2 floors | 17(11%) | 94(60%) | 45(29%) | 156 |
| Sum by damage level | | 81 | 221 | 142 | *444 |

**Table 4: Statistics of building damages caused by the earthquake. The percentage in the brackets was**

250 **calculated by the number in the cell divided by the total numbers of the row. *Sum of all affected buildings.**

Overall the significance in damage ratio could only be observed in damage level 1. There were relatively

more 1-floor buildings (22%) than 2-floor buildings (11%) survived. A difference related with

construction types was observed, as the survive rate of the RCM, WB, W types were 23%, 17%, and 9%.

There were only 4 RCF buildings and 2 survived. The damage ratios of the three major types (RCM, WB

and W), are shown in Fig. 4 A.

A damage pattern controlled by fault rupture was found. Building damage was more serious on the

hanging wall or within one-kilometer distance of the Yingxiu – Beichuan fault rupture (indicated by a

thick dotted line in Fig. 3). The ratio of buildings being destroyed in the northwest was much higher (Fig.

4 B and C) than in the southeast, as only 17 of the 81 survived buildings are located in the northwest. The

260 damage was not influenced by the construction types in the north, probably indicating the shaking was so

strong that it exceeded the resistance of all the three types (Fig. 4 B). The southern side showed a

significance difference in damage for the construction types, as the RCM buildings had the lowest

collapse ratio while wooden buildings had the highest (Fig. 4 C). The landslide area density in the

northwest is much higher than the southeast, further suggesting the existence of a localized ground

shaking difference (Fig. 3).

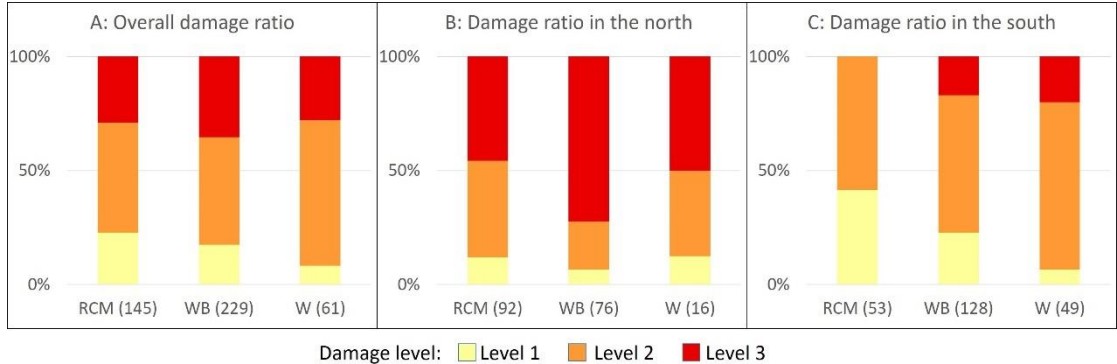

**Figure 4: Damage ratio statistics of the three major structural types in 2008. The numbers in brackets under the x axis indicate the total numbers of buildings. A: damage ratio of all the earthquake-affected buildings. B: damage ratio on the northern side of the dotted line in Fig. 5. C: damage ratio on the southern side of the dotted line in Fig. 3.**

Road stretches with a combined length of 3.7 km, which was 11% of the local road network of 33.5 km, were blocked by co-seismic landslides. The only access road, the tunnel in the southeast (Fig. 1 Access 1), survived the earthquake. None of the farmlands were directly affected by the co-seismic landslides, because most of them were located on gentle slopes or flat lands in the southern part.

**3.2 The disaster relief (2009)**

The aerial photos of 2008 and the SPOT image of 2009 were used to map shelters (Fig. 5). Before the government could bring in pre-fabricated houses the survivors set up 229 shelters by building shacks and using tents provided by the government. Many constructed the shelters next to their destroyed houses, even when this was very close to co-seismic landslides.

The government had problems in identifying suitable locations for the shelter settlements. The lack of awareness of the possible areas endangered by post-earthquake landslide and debris flow played an important role in this. Before the winter of 2008 four temporary settlements were made with 82 pre-fabricated buildings, which housed multiple families (Fig. 5 and Table 3). The largest temporary settlement with pre-fabricated buildings (PFM) along with some TSs was established on the lower part of the alluvial fan of one of the largest sub-watersheds, the Bayi catchment, which later posed a high debris flow threat, as 29% of its watershed area was covered by co-seismic landslides (Fig. 5).

It was difficult to estimate the accommodation status of the survivors since many of them went to relatives outside the area and many workers and soldiers stayed in the area to carry out the relief.

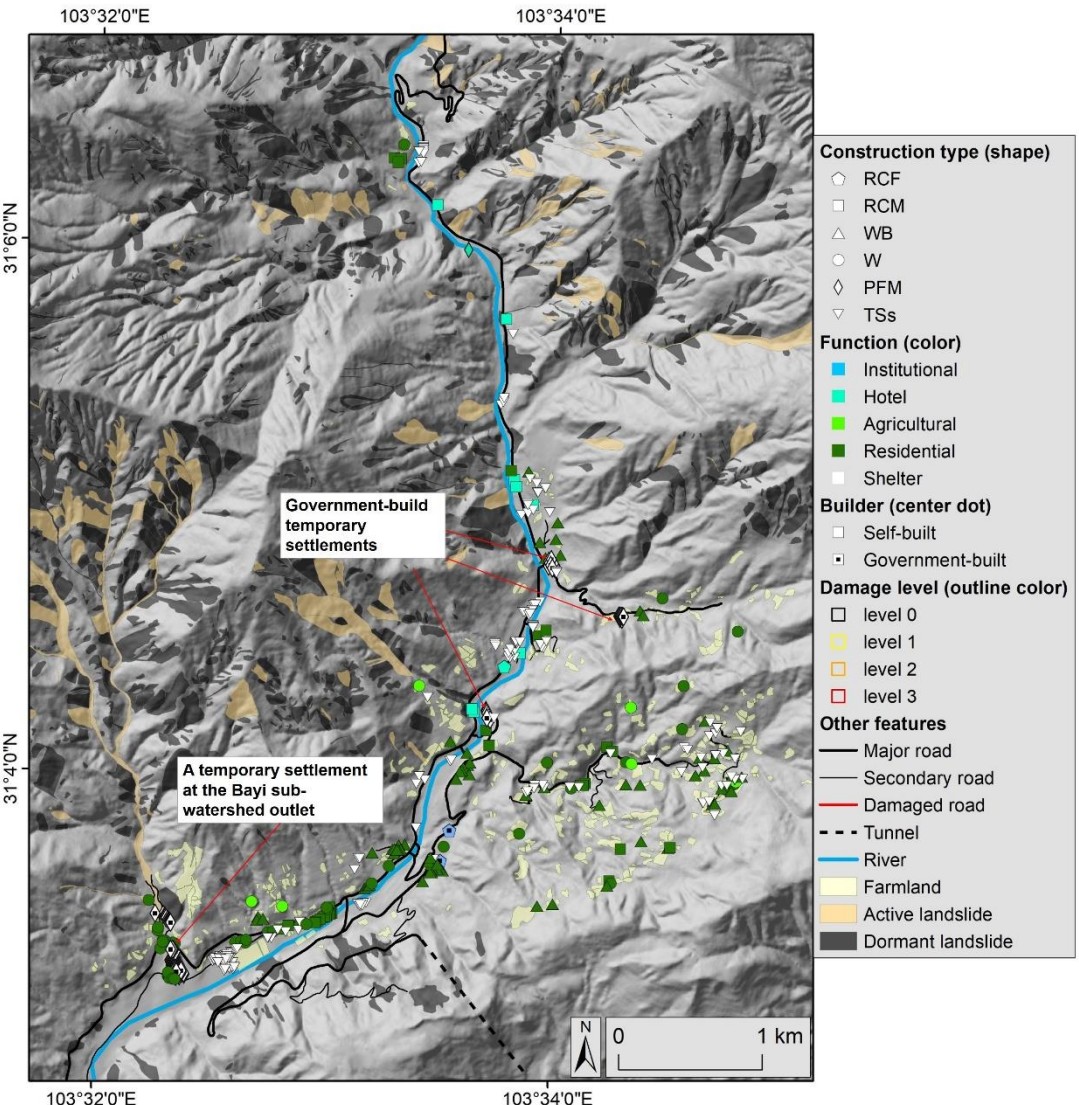

**Figure 5: Map of the buildings that survived the earthquake and location of temporary shelters. Many Local residences set TSs near their destroyed houses. The government established four PFM temporary settlements, and one of them was located at the outlet of the Bayi catchment, which posed a great debris flow threat due to a 29% landslide area density.**

**3.3 Early reconstruction stage (2009-2010)**

The SPOT image of 2009 and the Worldview-2 image of 2010 were used to map the buildings, roads, and mitigation measures for 2010, which illustrates the changes brought by early reconstruction efforts. The

city of Shanghai was assigned responsibility to execute the recovery activities of the nearby Dujiangyan city, and the surrounding area, including the Longchi valley. During this period all rubble was removed as well as most of the tents and shacks.

The new inventory contains 873 buildings, out of which 706 were newly constructed, including some new shelters. Among the 655 reconstructed permanent buildings, 481 were built by the residents themselves with the financial support from the government. There were 174 new buildings constructed by the government and most of them are concentrated in the center of Longchi town (Fig. 6), which was proven to be a safe location in the later years. All the road damages were repaired and a new highway

entrance was made in May 2009 (Fig. 1, access 2 and Fig. 8), which shortened the travel time to Longchi by nearly 40 minutes and bypassed some road sections threatened by landslides.

The government implemented a policy to avoid losses in future earthquakes and applied RCF structures for 99% of the reconstructed buildings. An example of such a government-built apartment building is shown in Fig. 2 A. The construction types for self-built residences did not change significantly, as most

of them (278) were built with locally available wood (WB and W construction types). A notable increase in using frame structures among the hostels was observed (Table 3), many of which were rebuild near the original locations along the Longxi River.

Unfortunately, the lack of knowledge about post-earthquake hazards had led to many careless decisions made by both the government and the local residents. Many buildings were rebuilt at outlets of

sub-catchments because historical deposition fans provided relatively flat land. Most of the rebuilt and newly added hotels are located next to the Longxi River in order to attract tourists, ignoring the potential danger posed by the river.

No major disaster occurred in 2009, as the precipitation was not significant in this region. Only limited hazard mitigation projects were carried out. Three potentially dangerous slopes near the Longchi town

were stabilized during the reconstruction process. After a small debris flow destroyed 9 PFM shelters during the monsoon of 2009, two check dams and a drainage were installed in the Bayi sub-watershed, (Fig. 6).

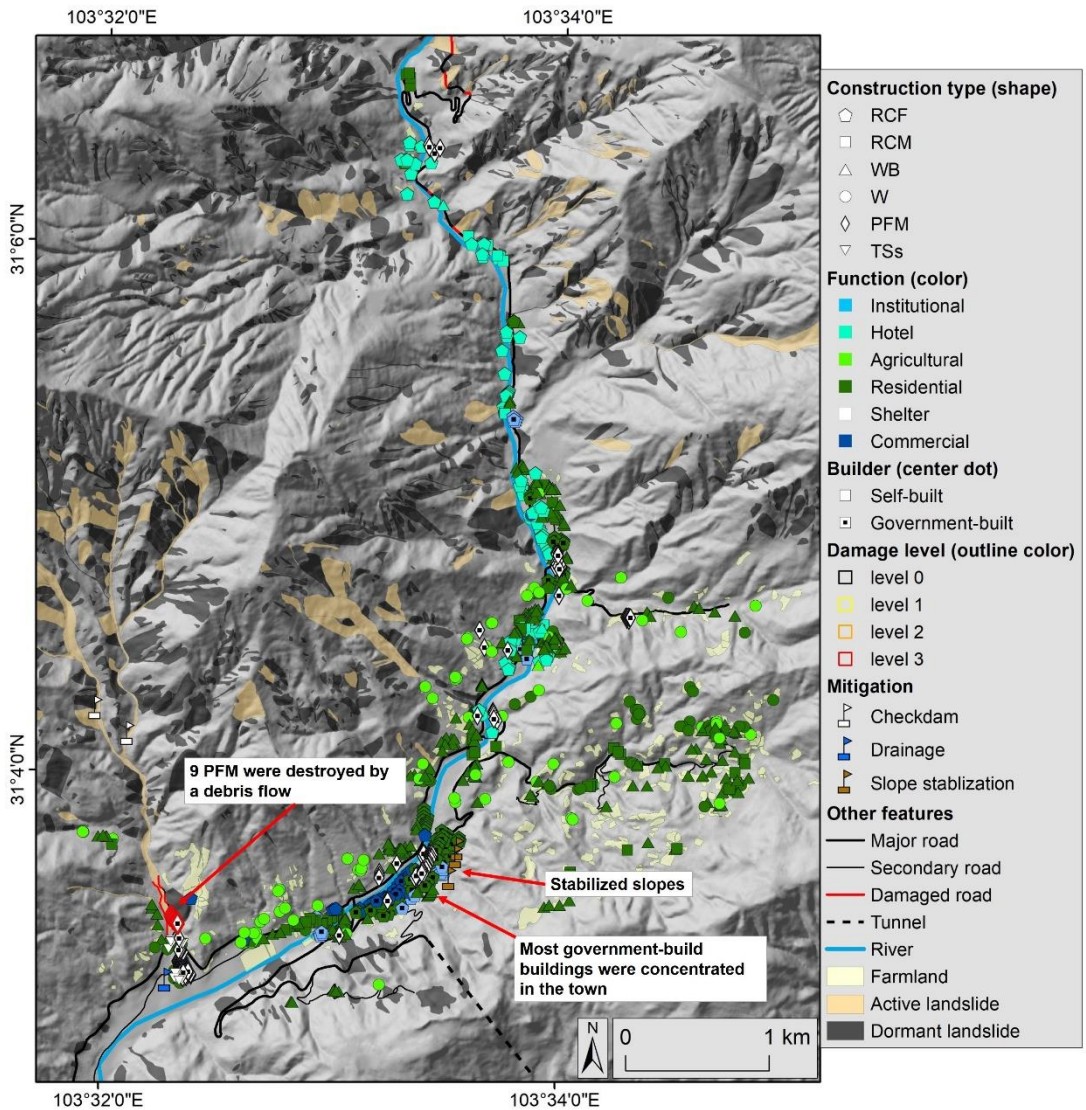

**Figure 6: The inventory of the situation in 2010 before the monsoon, showing the buildings, roads, and**

**remedial measures for the period between 2008 and 2010. Overlain are the active landslides in 2009. A debris**

**flow destroyed 9 PFM shelters, after which two check dams a drainage were installed.**

**3.4 The late reconstruction stage and major debris flow disaster (2011-2013)**

The Worldview-2 image from 2011 was used to map the changes caused by a major debris flow disaster

that occurred during 13 - 14 August 2010 (Fig. 7) (Xu et al., 2012;Tang et al., 2012). The event was

triggered by a storm on 14 Aug 2010 with a maximum recorded rainfall intensity of 75 mm/h measured

by rain gauges in the Longchi town (Xu et al., 2012). About 341 new landslides were triggered and 1151

of the co-seismic landslides were reactivated in this area during this event, producing several massive

debris flows which joined in the valley of the Longxi River (Yu et al., 2011), reaching the Zipingpu

reservoir lake. Sedimentation was 5 – 7 m at about 300 meters upstream of the town (Sichuan Geology

Engineering Reconnaissance Institute, 2011).

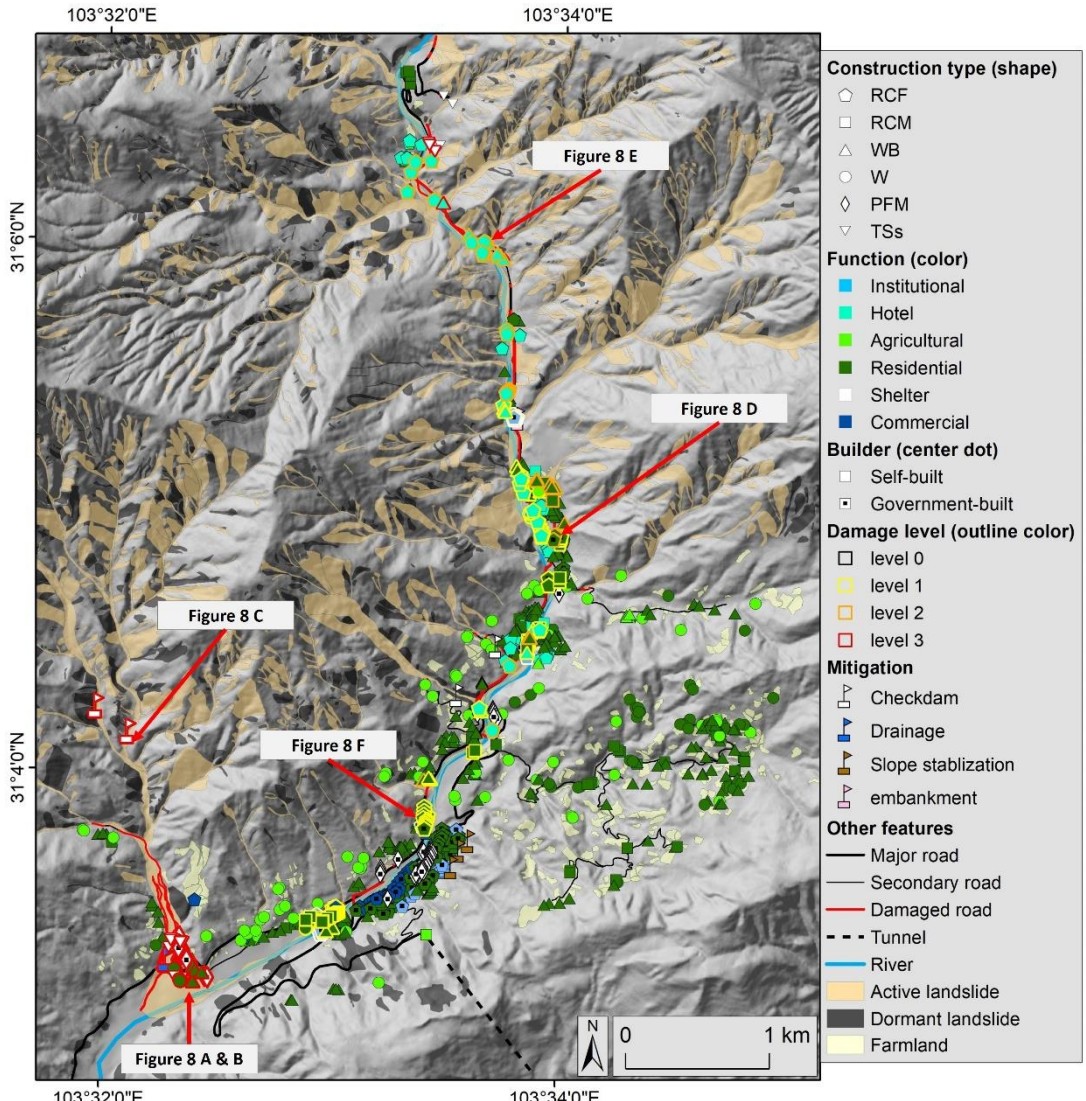

**Figure 7: The building and landslide inventories mapped based on a Worldview-2 image captured in April 2011, showing the changes brought by the August 2010 debris flow disaster. A total of 213 buildings were affected. The losses were largest for those buildings located either near sub-catchment outlets or close to the**

**river.**

Nearly one-fourth of all buildings in the study are were impacted by debris flows and subsequent floods.

Among all the 213 affected buildings, 70 were destroyed, 41 were severely damaged and 102 were

moderately damaged. The most severe loss occurred at the outlet of the Bayi sub-catchment, a large debris flow severely damaged the temporary settlement (Fig. 8 A & B). The drainage and the poorly

constructed check dams in Bayi sub-catchment constructed in the early 2010 did not prove to be adequate and were destroyed (Fig. 8 C). The disaster also damaged 35,000 m$^2$ of farmlands and destroyed 7.5 km of road. The losses were largest for those buildings located either near sub-catchment outlets (Fig. 8 A & D) or close to the river (Fig. 8 E & F).

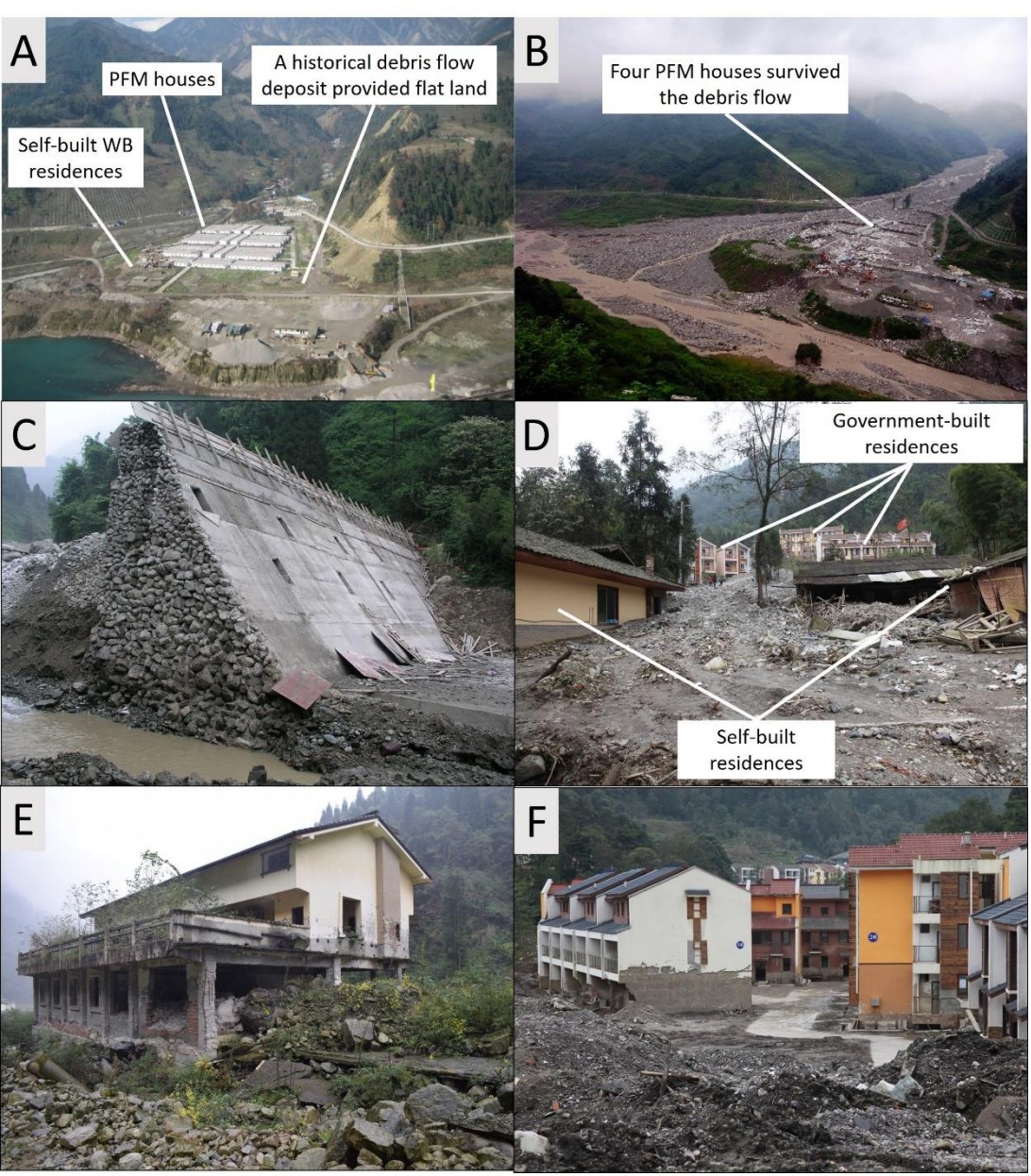

**Figure 8: Losses caused by the Aug 14, 2010 debris flows. The locations of the examples are shown in Fig. 8. A: The temporary settlement at the Bayi sub-catchment outlet in 2009 (Luo et al., 2010); B: The shelters**

**destroyed by a debris flow from the Bayi sub-catchment (Luo et al., 2010); C: One of the two under designed check dams in Bayi sub-catchment which were destroyed (Liu, 2010); D: Residences reconstructed on old debris flow deposits were damaged; E: A hotel beside the Longxi River was struck; F: government-built**

**apartment buildings beside the river were damaged.**

Another inventory was made based on the 2011 Worldview-2 and the 2013 Pleiades images , which represents the situation shortly after the debris flow and the official announcement of reconstruction completion (2012) (Fig. 9). All the temporary buildings were removed by 2012. A total of 38 buildings, that were threatened by debris flows or floods, were abandoned. The government constructed another 25

buildings to replace these and local people constructed 67 new buildings. The total numbers of functioning buildings were reduced to 678 (Table 3).

Many mitigation measures, such as check dams, sediment retention basins, and debris flow early warning systems, were implemented and concrete embankments were installed along parts of the river (Fig. 9). The debris flow warning is based on the accumulative rainfall and rainfall intensity recorded by rain

gauges installed in the watershed. A camera was installed in the upper stream of the Longxi river to monitor debris flow and flood activities.

From August 2010 to April 2013 the debris flow activities in most of the sub-catchments decayed rapidly except for the Bayi sub-catchment. A flashflood took place in 2013, damaging 20 buildings. A major cause of the floods was the dramatic raise of the riverbed (Yu et al., 2011) brought by debris flows. The

authorities said it was not possible to reopen the Longxi national park due to high landslide threat along the access road. The maintenance for the major road in the north stopped due to being repeatedly damaged by floods, and the closure of the national park. A dirt road was made as a replacement.

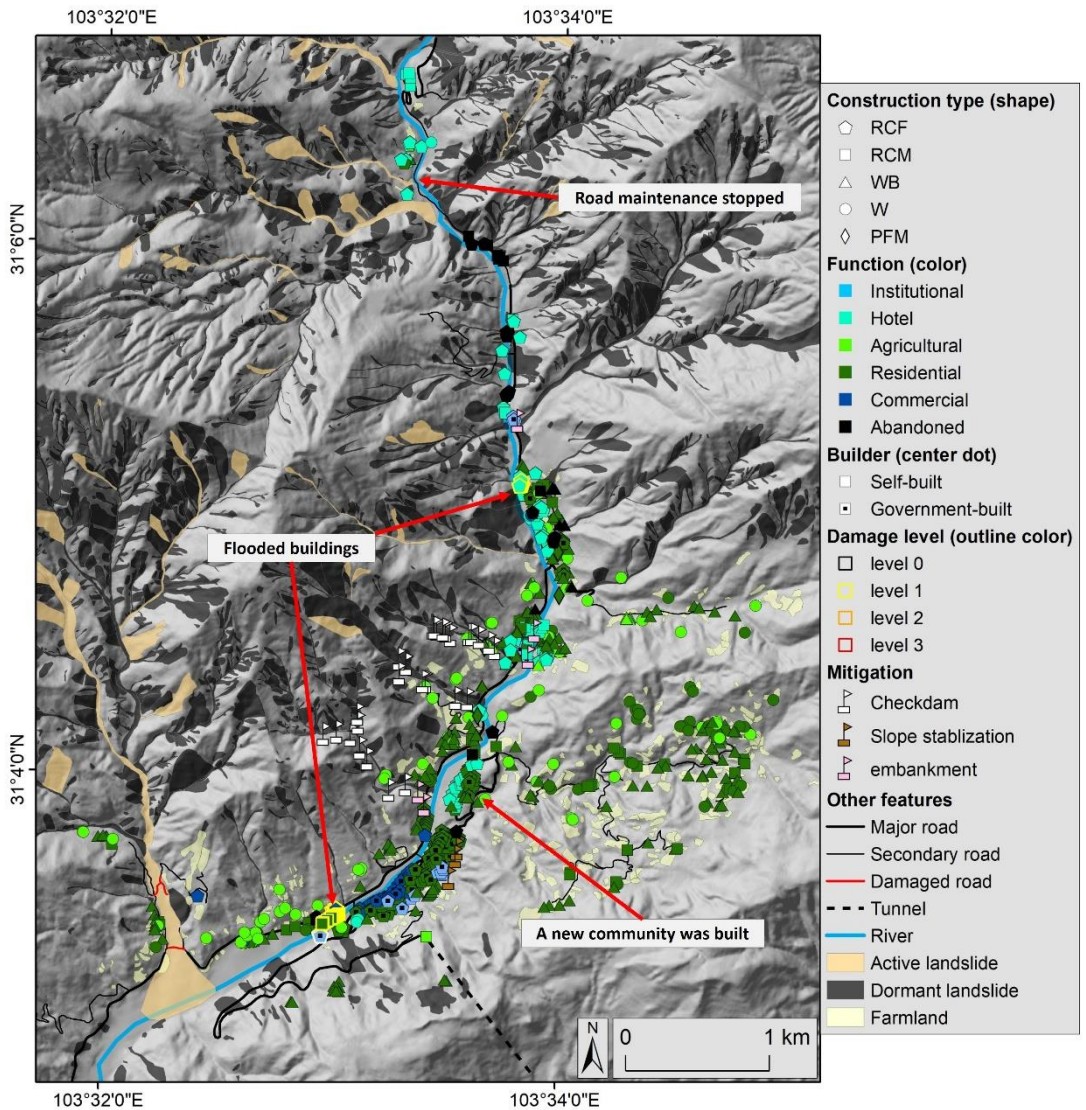

**Figure 9: The inventory of mapped based on the 2013 Pleiades image, showing the situation shortly after the official announcement of reconstruction completion. A new community was built to make up the loss caused by the 2010 debris flow and many mitigation measures were installed. The debris flows caused a rise of the riverbed and led to flooding.**

### 3.5 The post-reconstruction stage (2013 – 2018)

The changes from 2013 to 2018 was identified by interpreting the 2015 SPOT 6 image and the 2018 Pleiades image (Fig. 10). The society developed in a stable manner without any major disruption, thus we only described the inventory of 2018. In this period 21 new buildings were constructed by local people. The total number of buildings in the area grew to 699 (Table 3).

The road length increased to 46.2 km, as many dirt roads were made to access farmlands. The tunnel connecting the highway was closed due to water leakage (Fig. 1, access 2) and its maintenance was stopped, probably because of a low economic interest caused by the loss of tourism. Only the old tunnel (Fig. 1, access 1) could be used. A secondary road was made in 2018 connecting the neighboring catchment and provided a second access road for the Longxi watershed (Fig. 1 and Fig. 10, access 3).

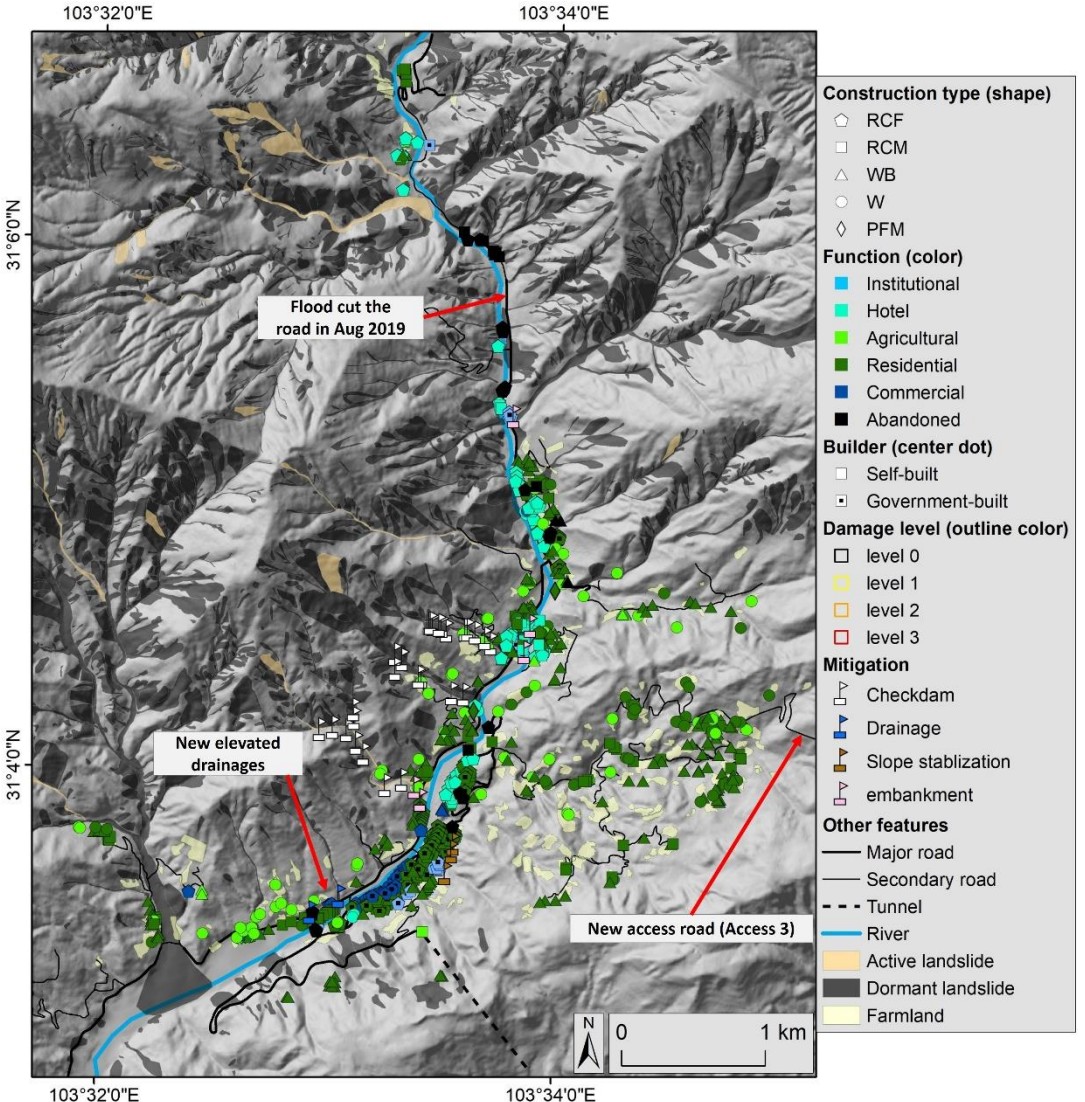

**Figure 10 The inventory made based on the 2018 Pleiades image, overlaying with landslide polygons made based on the standards of Tang et al. (2016). The society developed in a slow and stable manner, without major disruptions from 2013 to 2018. The last reported disaster was a flood cut the dirt road in the north on August 20, 2019.**

Landslides and floods did not cause any major loss since 2013 as due to the decaying hazard (Tang et al., 2016) and the mitigation measures. Two elevated drainage channels were installed in 2015 in the southern part to redirect flash floods from sub-catchments into the river directly. The last reported disaster was a flood cut the dirt road in the north on August 20, 2019.

## 4 Analysis of economic values

In this section the economic values of the built-up features were estimated in US dollar. The total value of the buildings was estimated by multiplying floor space with the unit price for construction. The values and the exposure in the seven investigated periods were evaluated.

### 4.1 Value estimation

The unit prices for different building types and roads were acquired through interviews with local builders and local government officers (Table 5). The unit prices of buildings increased after the earthquake due to several reasons: higher building standards, large consumption of building materials in the earthquake-hit areas, and currency devaluation. The price of mitigation structures were estimated based on the mitigation design of a catchment in the neighboring watershed (Li et al., 2011). The mitigation structures built after 2010 have a worth of approximately 30 million Yuan (Chengdu Bureau of Land and Resources, 2018). We were not able to acquire prices of farmlands, forests and other indirect factors. Therefore the analysis was limited to economic value, investment and direct loss caused by hazards. Severely damaged and destroyed buildings were counted as direct economic loss.

The economic value estimation result is illustrated in Fig. 11 A. As a result of the fast reconstruction, the total value increased rapidly to 96 million USD in 2010 and 133 million USD in 2013, which was nearly 5 and 7 times the value in 2007. This was caused by the increase in the number of buildings and the overall improvement in construction type, particularly the RCF buildings accounted for 75% of the total value.

The total direct loss during the monitored period was 16.5 million USD, out of which 8.4 Million was government losses and 8.1 million USD private losses. The disaster in August 2010 caused a loss of 8.3 million USD, which was slightly more than the loss caused by the Wenchuan earthquake. It is because

many expensive RCF buildings were carelessly built in areas exposed to debris flows and were severely

damaged beyond repair. The loss was further increased in 2013 in the form of buildings being abandoned

by the local residents in fear of debris flow and flood threat.

| Type | | Value | |
|---|---|---|---|
| Construction type | Code | Unit price before 2008 (USD / m²) | Unit price 2008 – 2012 (USD / m²) |
| RC frame structure | RCF | 217 | 415 |
| Reinforced masonry | RCM | 144 | 200 |
| Wood & brick | WB | 54 | 77 |
| Wooden | W | 27 | 46 |
| Pre-fabricated metal houses | PFM | - | 154 |
| Tents & shacks | TSs | - | 6 |
| Reinforced slopes | | - | *205 |
| Drainage channels | | - | *103 |
| Embankments | | - | *362 |
| Road (USD / m) | | | |
| Major road (6 m wide) | | 207 | |
| Secondary road (3 m wide) | | 23 | |
| Bridge (5 m wide) | | 828 | |
| Tunnel (6 m wide road) | | 5069 | |
| Others | | | |
| Mitigation works | | 4.6 million USD in total | |

**Table 5: Unit price of built-up features. All values were adjusted to the situation of 2012 by inflation rate of Chinese Yuan. *Calculated based on mitigation design of a nearby catchment.**

**4.2 Exposure**

The risk could only be expressed by the value of assets exposed to debris flows and floods, since we

could not quantify the return period of the highly dynamic post-earthquake hazards. The area affected by

the hazards were mapped based on the landslide inventories of Tang et al. (2016) and historical flood

traces found in field. Any building located in the affected areas was considered to have a potential

exposure (Fig. 11 B & C). A major increase in both of the value and number of the exposure in 2010

suggested a careless reconstruction plan. The decrease in 2011 was caused by the impact of the 2010

debris flow. After the debris flow the Longchi town adapted to the post-earthquake environment by

initiating multiple mitigation projects and invested 5 million USD (Fig. 11 B). By 2015 the majority of

the exposure were protected by mitigation structures.

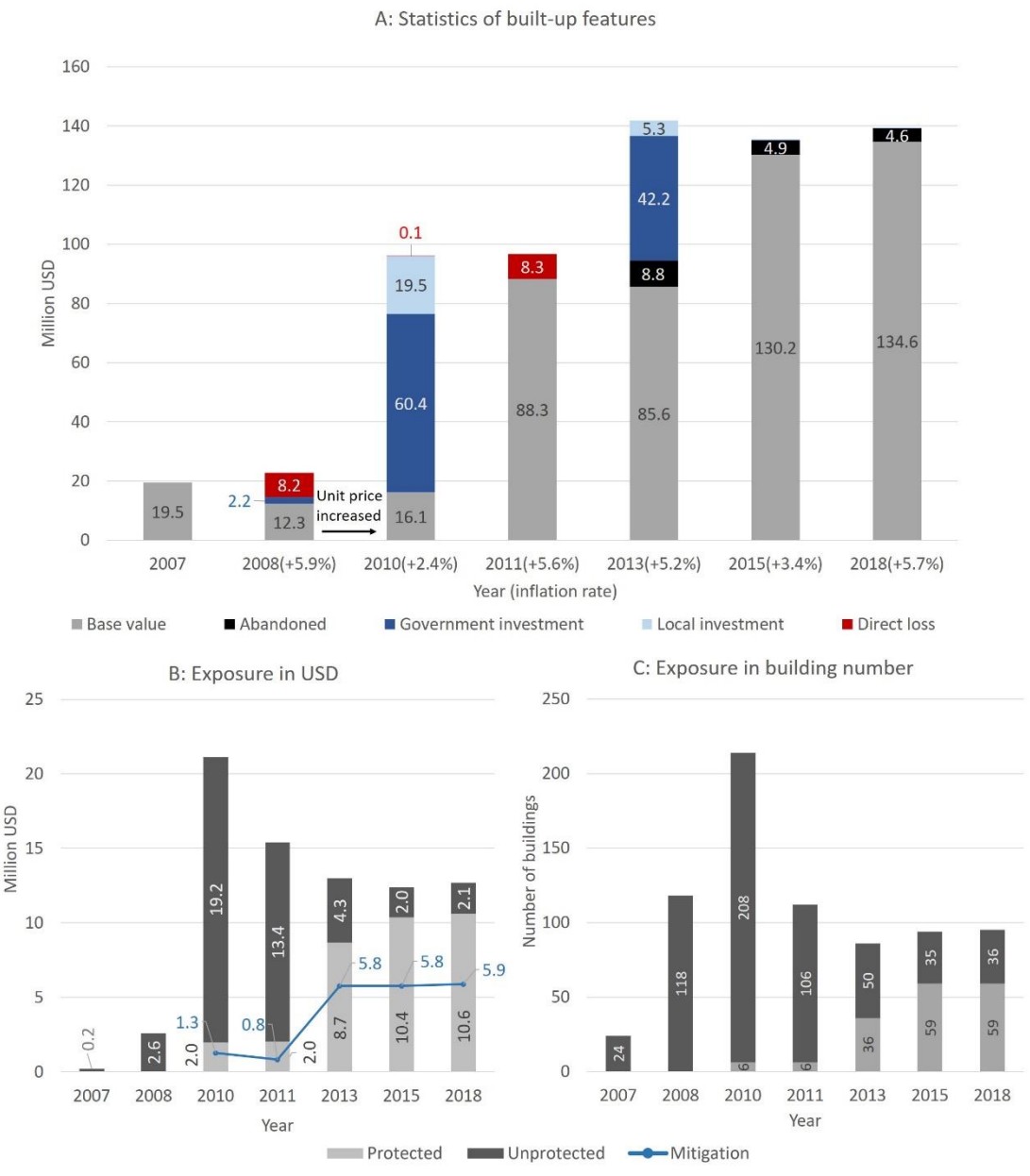

**Figure 11: A: the total values of the built-up features, investments and direct economic losses over the period**

**between 2007 and 2018 in the Longxi area. B: The total value of element-at-risk exposed to debris flows and**

**floods. The blue line indicates the amount of investment on mitigation measures. The values were adjusted with the inflation rate. C: The total number of buildings that exposed to debris flows and floods.**

**4.2 Economy**

The economy is described based on the interviews with the local residents and authorities. The economy prior to the earthquake relied mostly on farming, tourism, and working outside of the town. Forestry was an important economic activity in the heavily forested watershed of the Longchi River, with trees producing medicines, nuts and building materials. Agriculture and tourism were almost equally important, generating a gross output value of 6.9 million US dollar for the year of 1999 (Baidu

Encyclopedia, 2016).

After the earthquake the government distributed subsidies to the residents based on the reported property damage and organized several companies to employ the local people. There were 29% of the families completely or partially relied on working out side of the area in 2018, which was 9% higher than the pre-earthquake situation.

Tourism was stopped completely due to the earthquake, and only started recovering since 2015. Severe losses were taken after the reconstruction of hotels, as many were damaged by debris flows and floods, and they were unable to attract tourists due to the valley was considered as a dangerous place to visit. Although the business started flowing seven years after the earthquake, a fully recovery could only be expected upon the reopening of the national park.

The economy was more relied on agriculture than tourism after the earthquake. From 2007 to 2018, the farmlands have increased from 76 hectares to 98 hectares, and most of them are growing commercial crops. Sixty-five new agriculture buildings were built after the 2010 disaster, and many were used to house domestic animals such as chicken, ducks and goats.

**5 Discussion**

**5.1 Challenges faced in post-earthquake reconstruction**

Some existing examples such as Haiti (Jesselyn, 2017) and Nepal (Adhikari, 2017) are known for slow recovery process due to politics and limitations in economy. The Longchi town showed a contrary case

that rebuilding in a very rapid manner led to severe losses. The problems are majorly constructing many valuable assets in hazard-prone areas (spatial) and rebuilding too early before the environment could reach a relatively stable situation (time). The cause might be a lack of communication between the government and scientists because of the top-down political system. The necessity of hazard and risk assessment was not aware even though sharp increases in hazards after major earthquakes were reported before 2008 (Lin et al., 2004;Lin et al., 2006;Nakamura et al., 2000;Liu et al., 2013).

Ideally hazard maps should be updated shortly after earthquakes, considering the enhanced hazards and hazard chains (Fan et al., 2019a;Tang et al., 2016;Hovius et al., 2011;Marc et al., 2015), as well as the long-term dynamics. Upon acquiring hazard maps, multi-criteria analysis could be used to assess suitability for reconstruction planning, for example Barić et al. (2006) and Store and Kangas (2001). The possible consequences and potential risk created by the plans could be analyzed by land use models, for example Cammerer et al. (2012), and Promper et al. (2014).

The major difficulties of reconstruction planning may lie in hazard assessment, due to the spatial and temporal dynamics of hazards, as well as their interactions (Fuchs et al., 2012;Kappes et al., 2012). These difficulties are enlarged in a post-earthquake environment as many factors change much faster than they normally do. For example sediment discharge would be several times higher (Koi et al., 2008;Hovius et al., 2011) and vegetation regrew at a rapid speed (Yang et al., 2018;Liu et al., 2010). The commonly used evidence-based statistical assessment methods might be not valid shortly after earthquakes due to the changes in environment and triggering mechanism of hazards (Huang and Fan, 2013;Tang et al., 2011a;Tang et al., 2011b;Xu et al., 2012;Fan et al., 2019b). The application of deterministic methods would be more useful but is largely depending on the efficiency of data collection.

To our knowledge there are two models (van Asch et al., 2013;Bout et al., 2018) could simulate the both initiation and runout as well as incorporating temporal changes in environment, therefore have the potential to model such dynamics given sufficient data. The model of van Asch et al. (2013) simulates initiation and runout of entrainment-based debris flows and was tested in one of the sub-watershed in our study area. Bout et al. (2018) incorporated the functions proposed by van Asch et al. (2013) and the model allows the simulation of interactions of hazards (earthquakes, mass movements and floods),

further expanding its potential for post-earthquake hazard assessment. An example is given by Domènech et al. (2019) as they showed a case study to simulate the temporal changes in debris flow hazard affected by material depletion, revegetation and grain coarsening.

## 5.2 Exposure and mitigation

A sharp increase in exposure was demonstrated in this study, due to rises in both numbers and cost of

buildings. Although many element-at-risk were protected by mitigation measures against debris flows and floods or not affected, many self-built buildings are still vulnerable to earthquakes. It is hard to judge whether building more expensive RCF buildings than enough in such a highly earthquake-susceptible area is beneficial, for example buildings for commercial purposes. On one hand they potentially improved life quality for the community if the national park could be reopened in future, on the other this

increases the elements exposed to earthquake enormously in terms of economic value (Fig. 11 A). It is advised to keep a close track of the changes in elements-at-risk and hazard in order to understand the up-to-date risk situation. Automatic extractions could help this task if image data could be systematically collected and well geo-referenced.

Mitigation measures are widely used to reduce mass movement hazard (Fuchs et al., 2004;Keiler et al.,

2006;Hübl et al., 2005;Chen et al., 2015), however only when they are properly designed. A failed example was shown in this study (Fig. 8 C), as the magnitude of the 2010 debris flow exceeded the mitigation capacity. Such mitigation might create a false sense of security and possibly leading to more losses (Olugunorisa, 2009;Cigler, 2009). It might be not beneficial to start installing mitigation measures right after an earthquake, as the magnitude and frequency of mass movement hazards might be too costly

to mitigate. Several researches have shown that mass movement hazard activities decreased rapidly after 3-5 years (Fan et al., 2019a;Tang et al., 2016;Marc et al., 2015;Hovius et al., 2011), therefore a delay in mitigation and reconstruction could give more beneficial results than taking measures immediately after large earthquakes, in case of avoid building in risky areas is not possible.

## 6 Conclusions

We monitored the changes in the Longxi valley during a 11-year period after the Wenchuan earthquake and the subsequent recovery process, with seven inventories from different years containing buildings,

roads, land use and mitigation measures. Most of the stronger building construction types were only implemented after the earthquake, and mitigation structures were only installed after being impacted by debris flows and floods. A greater awareness to avoid living in hazard prone areas was observed after the

2010 debris flows. Despite the extensive and repeated damage, the earthquake, and subsequent landslides, debris flows, and floods gave Longchi town a chance to increase its resistance to these hazards in future, and to improve economically. Such is called development recovery (Davis and Alexander, 2016), which not only restoring all the recovery sectors but also to improve on what used to exist.

Due to the direct involvement of the Government and the city of Shanghai, who supported Longxi town financially and with expertise, the recovery was fast, considering the large loss and the mountainous terrain in the area impacted by the Wenchuan earthquake. The lack of experience of dealing with post-earthquake landslides was the largest flaw in the recovery planning. The damage caused by post-seismic landslides was not only restricted to Longxi, but was reported across the entire earthquake

affected region. The post-earthquake disasters did not significantly slow down the reconstruction process because of the strong economy of China, and the large amount of funding that was invested in reconstruction and protection using mitigation structures.

However, recovering the economy through tourism was a failure in Longchi town, because post-seismic debris flow activity was underestimated. Many resources were wasted, for example the destroyed and

abandoned hostels, the destroyed main road, and the revoked highway entrance. Similarly, despite of a handful of success examples, many unused and often destroyed tourism facilities can be seen all over the earthquake affected area. Among all the towns that had planned tourism, Longchi town had one of the worst failures, because its biggest attraction was the national park which could not be reopened. The recovery would have been much more efficient if it included the awareness of risk management.

However, the question remains if these hazard reactivations could have been predicted and mitigated properly.

In such a mountainous region it is recommended not to re-build near the outlet of catchments containing many co-seismic landslides. Limiting reconstruction too close to rivers is also recommended to avoid

floods caused by riverbed raising and landslide dams. Avoiding build critical structures and residential

buildings near major faults, like the Yingxiu – Beichuan fault in Fig. 3, could lower the risk posed by

earthquakes. The exact areas susceptible to hazards should be acquired by conducting hazard

assessments.

A possible timetable for recovery actions is presented in Table 6. Four post-seismic phases are identified

based on landslide activity from Tang et al. (2016) and Fan et al. (2018). Period I (very high) means the

period when the majority of co-seismic material in streams is not depleted and loosened slope materials

have not failed. Period II (High and fast decay) means the time after the first major mass movement event

which removes the majority of the stream blockages. Landslides occur frequently but not likely with a

catastrophic magnitude in this phase. Period III (Low and slow decay) is when vegetation has mostly

recovered and landslide activity is no longer frequently observed. Landslide activity is much lower

compared with the previous two phases, and slowly decays towards the pre-earthquake level, but still

pose a significant threat. Period IV (Fully recovered) is when landslide activity is at the same level as the

pre-earthquake rate.

| Post-seismic landslide threat | Very high | High and fast decay | Low and slow decay | Fully recovered |
|---|---|---|---|---|
| Period number | I | II | III | IV |
| In the Longxi watershed | 2008 - 2010 | 2010 - 2015 | 2015 - ? | Unknown |
| Rebuilding in risky zones with mitigation | 0 | 0 | 1 | 2 |
| Rebuilding in safe zones | 1 | 2 | 2 | 2 |
| Installing mitigation works | 0 | 1 | 2 | 1 |
| Reopen tourism | 0 | 0 | 1 | 2 |
| Hazard surveys frequency | Very High | High | Moderate | Same as pre-earthquake |

**Table 6: Suitability of actions after earthquakes, using the Longxi watershed as the example. 0 = not suitable, 1 = somewhat suitable, 2 = suitable**


In period I rebuilding should be strictly limited to areas with low disaster threat. Even then risk still exists since rivers could be dammed by mass movement and cause flooding in areas outside of landslide-prone zone. It might not be appropriate to install mitigation works unless it is absolutely necessary because of expensive cost and high magnitude of hazards. Hazards, particularly mass movements, should be closely monitored in order to respond to emergencies in a timely manner. In period II extreme disasters are less likely to occur but building in landslide-prone area is still too risky. Mitigation works could be installed in key locations to keep critical infrastructures. In non-critical locations it is not beneficial to install mitigation works yet due to large amount of co-seismic debris that could still be easily activated by rainfall. It is still necessary to closely monitor hazards. Period III is the optimal time to install mitigation measures since the mass movement threat would be relatively low, therefore easier to be controlled. Buildings are allowed to be constructed in risky areas under the protection of mitigations in order to utilize the limited space in a mountainous area. Reopening tourism is possible during dry seasons but should be limited during wet season, which could be periods of monsoon, tropical cyclone and ice melting. In period IV the environment is fully recovered and construction plans and hazard survey could be carried out as the pre-earthquake situation.

It could be concluded that hazard and risk assessment are necessary for a properly conducted post-earthquake recovery. The assessments should consider not only spatial but also temporal dynamics of hazards, as well as the possible interaction among different hazard types, so that proper locations and time for reconstruction could be acquired.

**Data availability.** The multi-temporal landslide and element-at-risk inventories are available at https://www.researchgate.net/profile/Chenxiao_Tang2.

**Author contributions.** This work was carried out by Chenxiao Tang as part of his PHD thesis under the supervision of CVW. The field investigation and mapping were carried out by Chenxiao Tang, XL, YC, YY, HT, and CY. Chuan Tang provided resources and data.

**Competing interests.** The authors declare that they have no conflict of interest.

**Acknowledgement.** This research was supported by National Key Research and Development Program of China (2017YFC1501004) and National Natural Science Foundation of China (41672299). We would like to thank two anonymous reviewers for their helpful comments and suggestions.

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

845

**Appendix A**

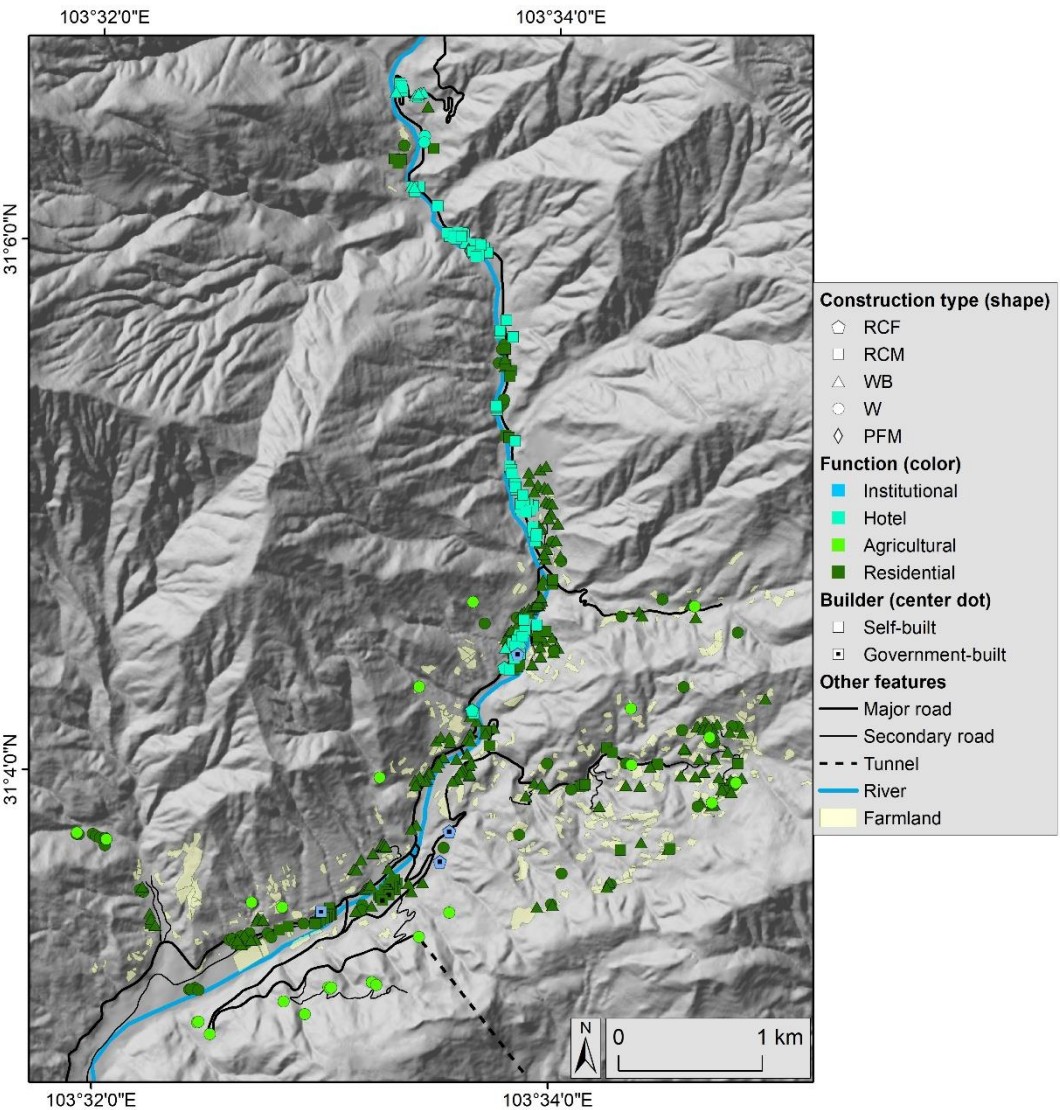

**Figure A1: A figure showing the situation in 2007, before the Wenchuan earthquake. Most buildings were self-built by local residents and many adopted WB and W construction types. Most of the region were densely covered by vegetation and no active landslide was observed.**