# Peer review of "Monitoring of the reconstruction process in a high mountainous area affected by a major earthquake and subsequent hazards"

_Natural Hazards and Earth System Sciences, 2019_

## Referee Comment (RC1) · Anonymous Referee #1 · 25 Oct 2019

General comments:

The paper describes the monitoring the post-earthquake reconstruction process of a town in China over 11 years. The authors use a range of remote sensing data from different times for image interpretation complemented with field mapping, interviews and literature surveys. Additionally, the manuscript provides an analysis of economic values. It seems that a lot of effort and time were needed to perform the analyses. The manuscript is well prepared and the provided information interesting. However, it reads rather like a report and not like a scientific research paper and the level of innovation regarding the methods is low. This is my major concern and makes it also difficult

for me to judge, if the paper should be accepted or not, even if the topic itself fits for NHESS. Also, parts of the descriptions are a bit lengthy, for example the descriptions of the economic values (even if relevant).

Specific comments:

Line 68: "The increased debris flow activity lasted for five years...". Do you have an explanation for that (maybe I missed it)?

Line 93: This is repetitive to the previous sentence.

Section 1.3: Is all the description relevant for the paper? It is a bit long.

Line 176 ff and in general: Did the authors consider the usage of any automated change detection approaches for image analysis? For some classes this might have been helpful and faster than digitizing all the features. For example, there are several studies and publications that successfully used such methods for post-earthquake damage assessment.

Figure 4: The color of the dormant landslides is not very well visible (it is better in the following figures). What is the dashed line in the image? There is only information later in the text, but not in the legend or the caption.

Line 265: First sentence. What is the reason for that?

Technical corrections:

In general, the paper is well written. However, spell check is needed, several typing errors need to be improved, partly formatting (chapter 1.3) should be adapted.

Table 4: The text "Sum..." is not readable.

---

## Author Comment (AC1) · 7 Nov 2019

Dear referee 1

Thank you for your comments. I will reply your comments on behalf of all the authors of this manuscript.

I agree there is not much innovation in the method, as they are just mapping and statistics, though the result was interesting. I believe science is not only about innovation in method, but also discovering new knowledge and passing it to the society. Our manuscript has shown the necessity of careful planning when recovering from a major

earthquake, instead of rushing to reconstruction as China did. We hope this work will provide knowledge to international communities that are threatened by earthquakes and post-earthquake hazards. From the aspect of sciences this is part of our plan towards the risk quantification method framework in post-earthquake environment, which is poorly studied.

Sincerely

Chenxiao Tang

————————————————————————————————————————

Specific comments:

————————————————————————————————————————

Your comment: Line 68: "The increased debris flow activity lasted for five years". Do you have an explanation for that (maybe I missed it)?

Reply: The sentence means: The Wenchuan earthquake created large amount of mass wasting and loss of vegetation, which amplified the debris flow activities. As this part was not well-explained, I deleted "The increased debris flow activity lasted for five years" and merged it the upper paragraph. The original lower paragraph was revised (removed descriptions about vegetation recover as it is not so relevant) to explain the enhanced landslides after the earthquake:

The catastrophic debris flows were the result of landslide activities amplified by the destabilized environment. In the epicentral area of the 2008 Wenchuan earthquake, the total active landslides was and has decreased largely in the first five to eight years (Tang et al., 2016;Yang et al., 2017;Yang et al., 2018;Zhang et al., 2016). Similar recovery patterns of co-seismic landslide surface were also observed In the Mianyuanhe area of the Wenchuan earthquake affected region (Li et al., 2016). On Aug 20 2019, several debris flows severely damaged the reconstructed settlements and roads in the Wenchuan area.

——————————————————————————————————————————

Your comment: Line 93: This is repetitive to the previous sentence.

Reply: The paragraph (line 90 - 95) was revised as: In this study we generated seven inventories of elements-at-risk covering a period of 13 years (2005 - 2018) to study the recovery of Longchi valley, located close to the epicenter of the Wenchuan earthquake. Image interpretation was carried out based on a series of satellite images collected between 2005 and 2018 and several field surveys were conducted. The study aims to demonstrate and analyze the process of post-disaster recovery in an unstable geo-environment disrupted by a major earthquake.

——————————————————————————————————————————

Your comment: Section 1.3: Is all the description relevant for the paper? It is a bit long.

Reply: We tried to shortening the section by removing some not so relevant sentences.

——————————————————————————————————————————

Your comment: Line 176 ff and in general: Did the authors consider the usage of any automated change detection approaches for image analysis? For some classes this might have been helpful and faster than digitizing all the features. For example, there are several studies and publications that successfully used such methods for post-earthquake damage assessment.

Reply: We considered using automated method. Due to the limited quality of our data (e.g. Spot 5 image), large areas of bare surfaces created by landslides, vegetation that expanded above the houses, and dusts created by the earthquake in the 2008 image, simple classification methods could not extract the buildings. We do not have more data and advanced commercial software for OOA analysis, thus our only option was by manual digitization. It is also the easiest way to ensure temporal consistency among the multi-temporal inventories.

Shortly after we finished the first version of the element-at-risk inventory we acquired a Landover map of 2014 from the government. We carefully checked and edited our data based on the official map to minimize error.
* * *
Your comment: Figure 4: The color of the dormant landslides is not very well visible (it is better in the following figures). What is the dashed line in the image? There is only information later in the text, but not in the legend or the caption.

Reply: In Figure 4 and 6, there were no dormant landslides because all of them were freshly triggered. I removed dormant landslides from the legend in these two figures and added the description of the thick dashed line in the caption.
* * *
Your comment: Line 265: First sentence. What is the reason for that? It is caused by large numbers of the destroyed 1-floor WB buildings, which itself was more than all 2-floor buildings combined. This was a careless description.

Reply: This sentence is rewritten: Overall the significance in damage ratio could only be observed in damage level 1. There were relatively more 1-floor buildings survived (22%) than 2-floor buildings (11%). This pattern could be observed from all building types. A difference related with different construction types was observed, as the survive rate of the RCM, WB, W types were 23%, 17%, and 9%. There were only 4 RCF buildings and half of them were repairable. The damage ratios of the three major types (RCM, WB and W), are shown in Figure 5 A.
* * *
Technical corrections:
* * *
Your comment: In general, the paper is well written. However, spell check is needed,

several typing errors need to be improved, partly formatting (chapter 1.3) should be adapted.

Reply: We did spell and format check.
* * *
Your comment: Table 4: The text "Sum: : :" is not readable.

Reply: This was caused by an error when converting .docx file to .pdf file. I adjusted the line spacing of the table to make it visible.

---

## Referee Comment (RC2) · Anonymous Referee #2 · 18 Nov 2019

The authors present a multi-temporal study on an area near Chengdu, China, affected by earthquakes and subsequent gravitational mass movements such as landslides of different types and debris flow hazards. The main aim of the work is related to dynamics in elements at risk exposed, and the economic impacts caused by a hazard chain (from earthquake to mass movements). The authors show how planned reconstruction may result in high loss in cases were relocation places are not chosen in an optimal way, and how subsequent hazards following a major hazard as trigger may affect the overall risk in the case study region.

The overall topic is of considerable interest to the readers of NHESS as it is one of

only few studies related to a quantification of hazard chains and underlying dynamics, including values at risk. As such, it would be good to see such material published, nevertheless, there are major shortcomings that need to be addressed before the material may become acceptable for publication.

The following general items should be addressed:

- This study on the effects of earthquakes and subsequent other hazards is an interesting case study, but authors do not tell potential readers what is novel. This needs to be done both at the beginning but also in discussion, telling potential readers the importance of multi-hazard and risk studies, such as e.g. done by Kappes et al. (2010; 2012a; 2012b). Moreover, the overall literature provided is quite restricted to Chinese sources while in the international field many other studies exist on distinct aspects, such as hazard and risk chains (see Kappes above), hazard and risk dynamics (Fuchs et al., 2013) and general land use dynamics (Cammerer et al., 2013; Rougé et al., 2015; Fuchs and Glade, 2016), etc. So it is highly recommended to extent the review to the broader context of such international sources, which in turn would allow the international readers to better understand the situation in China.

- Moreover, if authors were to explain the results of their case study to someone in another country, what would they gain from this Chinese case study? Do they learn from the methodology applied?

- In many phrases, authors provide facts, information, or ideas, but without supporting sources as to where those ideas or facts came from. All facts and information that are not common knowledge need to have citations (which are then put into the reference list) and all items in the reference list should be cited at least once in the manuscript. Examples include but are not limited to: Section 1.1., lines 30-36 and 40, then further lines 270-284, and the events descriptions (e.g., in section 3.4) and also the final section 5.

- The niche and gap for this research have to be clearly addressed in section 1.1 so

that the overall need and motivation for this work becomes clear, ideally this will go in lines 85-90 of the present manuscript.

- Section 1.3 has to be shortened and included in section 1.2

- Figures: fonts are too small, Figures need north arrow, measured grid and maybe an inlet showing the case study area in China.

- Figure 2: only Figure 2A is mentioned in the main text body.

- Please carefully check the English again, even if the material reads fluently, there are some minor errors such as debrisflow versus debris flow, etc.

- As already noted by referee #1, the overall text reads more like a technical report as a scientific paper. As such, I recommend to shorten the overall text by putting more information in Tables, and by combining some of the materials presented to that the overall appearance becomes more concise.

I encourage the authors undertaking major revisions so that the material becomes more accessible because according to my opinion the overall study is definitely worth being published and discussed with the broader scientific community. Recovery from disasters is an asset, and should get broader attention (Davis and Alexander, 2016).

References mentioned

Cammerer, H., Thieken, A. H., and Verburg, P. H.: Spatio-temporal dynamics in the flood exposure due to land use changes in the Alpine Lech Valley in Tyrol (Austria), Natural Hazards, 68, 1243-1270, 2013.

Davis, I., and Alexander, D.: Recovery from disaster, Routledge, London, 357 pp., 2016.

Fuchs, S., Keiler, M., Sokratov, S. A., and Shnyparkov, A.: Spatiotemporal dynamics: the need for an innovative approach in mountain hazard risk management, Natural Hazards, 68, 1217-1241, https://doi.org/10.1007/s11069-012-0508-7, 2013.

Fuchs, S., and Glade, T.: Vulnerability assessment in natural hazard risk - a dynamic perspective, Natural Hazards, 82, 1-5, https://doi.org/10.1007/s11069-016-2289-x, 2016.

Kappes, M., Keiler, M., and Glade, T.: From Single- to Multi-Hazard Risk Analyses: a concept addressing emerging challenges, in: Mountain risks: bringing science to society, edited by: Malet, J.-P., Glade, T., and Casagli, N., CERG, Strasbourg, 351-356, 2010.

Kappes, M., Keiler, M., von Elverfeldt, K., and Glade, T.: Challenges of analyzing multi-hazard risk: a review, Natural Hazards, 64, 1925-1958, https://doi.org/10.1007/s11069-012-0294-2, 2012a.

Kappes, M., Papathoma-Köhle, M., and Keiler, M.: Assessing physical vulnerability for multi-hazards using an indicator-based methodology, Applied Geography, 32, 577-590, https://doi.org/10.1016/j.apgeog.2011.07.002, 2012b.

Rougé, C., Mathias, J.-D., and Deffuant, G.: Vulnerability: From the conceptual to the operational using a dynamical system perspective, Environmental Modelling and Software, 73, 218-230, 2015.

---

## Author Comment (AC2) · 18 Dec 2019

Dear referee 2

Thank you for your comments and useful literature recommendations. I will reply your comments on behalf of all the authors of this manuscript.

I appreciate your approval for the value of the work. I found that not much attention has been paid on the aspect of post-disaster recovery – how, when and where should recovery being carried out and how prolonged effect from earthquakes would affect communities. I considered this manuscript would be an interesting case to show the

importance of having a good understanding of hazard and careful planning after major earthquakes.

I agree that some parts of the manuscript were poorly written and did not show the significance and value of the topic. Therefore, I have rewritten many texts, which are majorly in the introduction and conclusion sections.

Sincerely

Chenxiao Tang

Specific comments

Your comment: This study on the effects of earthquakes and subsequent other hazards is an interesting case study, but authors do not tell potential readers what is novel. This needs to be done both at the beginning but also in discussion, telling potential readers the importance of multi-hazard and risk studies, such as e.g. done by Kappes et al. (2010; 2012a; 2012b). Moreover, the overall literature provided is quite restricted to Chinese sources while in the international field many other studies exist on distinct aspects, such as hazard and risk chains (see Kappes above), hazard and risk dynamics (Fuchs et al., 2013) and general land use dynamics (Cammerer et al., 2013; Rougé et al., 2015; Fuchs and Glade, 2016), etc. So it is highly recommended to extent the review to the broader context of such international sources, which in turn would allow the international readers to better understand the situation in China.

Reply: Major revision was done for the introduction and conclusion sections. Many new texts and re-structure were made. A separate section of recommendation was made after the conclusion, to fulfill the goal stated in the introduction: to fill the knowledge gap about how, when and where to rebuild after major earthquakes.

Your comment: Moreover, if authors were to explain the results of their case study to someone in another country, what would they gain from this Chinese case study? Do they learn from the methodology applied?

Reply: A recommendation section was added to address this issue.

Your comment: In many phrases, authors provide facts, information, or ideas, but without supporting sources as to where those ideas or facts came from. All facts and information that are not common knowledge need to have citations (which are then put into the reference list) and all items in the reference list should be cited at least once in the manuscript. Examples include but are not limited to: Section 1.1., lines 30-36 and 40, then further lines 270-284, and the events descriptions (e.g., in section 3.4) and also the final section 5.

Reply: References were added to these descriptions. In where literatures could not be found, I rephrased them to show they are not certain conclusions.

Your comment: The niche and gap for this research have to be clearly addressed in section 1.1 so that the overall need and motivation for this work becomes clear, ideally this will go in lines 85-90 of the present manuscript.

Reply: a major revision of the introduction section was carried out. Most texts were rewritten.

Your comment: Section 1.3 has to be shortened and included in section 1.2

Reply: We shortened and merged it with section 1.2 based on the comments of you and referee 1.

Your comment: Figures: fonts are too small, Figures need north arrow, measured grid and maybe an inlet showing the case study area in China.

Reply: Most of the figures are designed to be printed in full or nearly full page size to make elements visible and the font size was designed to fit for that. North arrows have been added in all maps. A grid and an inlet map have been added in Figure 1.

Your comment: Figure 2: only Figure 2A is mentioned in the main text body.

Reply: The description of construction types was accidentally removed before submittion. I wrote a new paragraph that describes construction types briefly and referred to the figures.

Your comment: Please carefully check the English again, even if the material reads fluently, there are some minor errors such as debrisflow versus debris flow, etc.

Reply: I corrected several spelling and grammar mistakes.

Your comment: As already noted by referee #1, the overall text reads more like a technical report as a scientific paper. As such, I recommend to shorten the overall text by putting more information in Tables, and by combining some of the materials presented to that the overall appearance becomes more concise.

Reply: I tried to remove some non-essential part. In my opinion it would be nice to tell the story in a detailed manner than summarization that common researches would do, as the descriptions may provide information to planners and engineers. If the referees insist that a shortened version will be better, then the introduction about the Wenchuan earthquake and monitoring sections could be largely shortened in the next revision.
* * *
**Legend**

- ◎ Town location before 1998
- ◉ Town location 1998 - 2008
- ⦿ Current town location
- ▬ Highway bridge
- — Major road
- — Secondary / dirt road
- ▪▪▪▪ Highway tunnel
- ---- Tunnel
- — Ruptured fault
- — River
- ▨ Lake
- ▨ National park
- ▨ Longxi watershed
- ▨ Buildings (2018)
- ▨ Zipingpu Hydropower Dam
- ▭ Study area

Longchi Lake

Longxi River

Yingxiu – Beichuan Fault

Guanxian – Jiangyou Fault

Minjiang River

Access 1 (Tunnel)

Access 2 (Tunnel to highway)

Access 3 (New road in 2018)

To Dujiangyan

0          3 km

**Fig. 1.**

---

## Author Response (AR3)

**Comments**

*Comment from editor*

**Firstly, I kindly ask you to consider ALL of the reviewer comments with the same rigour. Both of the referees argued that the manuscript needs a revision with respect to the overall accessibility and reading fluency. This specifically includes the introductory section (section 1.1), as mentioned in both of the comments of referee #2. The current version, however, still lacks in organisation and proper referencing.**

Reply: Now all the comments from the referees are addressed. I apologize for the previous revision as it was indeed not satisfying. I had problems to access many materials due to a change in my working place. Now all maps were remade from scratch. I also tried to rewrite many parts of the manuscript and added a new discussion section.

**Secondly, lots of the material presented is not supported by either references or an indication whether or not it is a result of your own studies, examples include section 3 (from where exactly do you see that the reconstructed houses are governmental?). Also, to give another example, information in Line 141 is fragmentary: "The earthquake triggered a total of 1597 landslides in the watershed, which crashed four hostels, killing ten." Here information is missing: "resulting in ten lives lost" or similar could be an alternative sentence; moreover, it is not clear from which source the number of landslides comes from. Moreover, information such as the "national park was closed due to high landslide threat" need proper referencing. Further, "In May 2009 the 7.3 km long Longxi tunnel (Figure 1) was completed for the Duwen Highway, which greatly helped disaster relief and reconstruction by reducing travel time greatly" – apart from the grammatical issues, which is the source here to prove that this road section "greatly helped" disaster relief and reconstruction? These examples (and there are many more in this manuscript) illustrate nicely what was originally mentioned by the referees as the manuscript needs to be re-organised and "in many phrases, authors provide facts, information, or ideas, but without supporting sources as to where those ideas or facts came from". Please carefully solve this issue!!!**

Reply: Many descriptions in section 2 and 3 were from field investigation and interviewing local authorities. Fragmented sentences were rewritten, and the source of claims (builders and the national park status) were described as by interviews. Descriptions that we did not have solid proof were removed, such as "…which greatly helped disaster relief and reconstruction by reducing travel time greatly"

**Thirdly, if referees kindly ask you to add information to Figures, such as the mentioned measured grid and the north arrow, why did you not follow this request throughout the entire manuscript? Your figure captions and table headers are exceptionally short and need to be made such that your figures and tables are self-standing. In other words, if a reader were to read the figure caption and the table header, look at the figure and the table, but not have the rest of the text from the paper, would they be able to understand what is written?**

Reply: These figures are now remade and the captions are extended.

**Fourthly, as indicated by referee #1, the manuscript would gain in a discussion section where most of the findings are mirrored against the material from other studies published (the international literature requested by the referees) – here it is also highly recommended not only to follow those works exemplified by the referees but also to search own ones (and to my knowledge there are many available even with one of the co-authors as contributor), please carefully re-check available literature.**

Reply: A new discussion section is added.

**Finally, I kindly would like to ask you for a proof reading of your manuscript – at the moment, many sentences are not written in a style formal enough to be suitable for a scientific journal like NHESS. I am not saying the English must be grammatically perfect, but at least to a level that the reviewers (and myself) can understand what is being said scientifically. Moreover, you are kindly**

**requested to carefully consider the author guidelines when preparing your manuscript, this also includes the reference list which is NOT yet formatted according to the NHESS style.**

Reply: Many sentences are now rephrased. NHESS style is now applied in the references.

**Once again, please take these concerns serious. I am looking forward to receiving your revised piece of work as soon as possible. Should there any questions arise please feel free to contact me at any time, and please assure that all the co-authors have seen and approved the new manuscript version before re-submission.**

**Kind regards,**

**Sven Fuchs (Editor NHESS)**

*Comments of referee 1*

**Your comment: Line 68: "The increased debris flow activity lasted for five years". Do you have an explanation for that (maybe I missed it)?**

The sentence means: earthquakes create large amount of mass wasting and loss of vegetation, which amplified the subsequent mass movement activities. The amplification decays as environment recovers.

The introduction was re-written to make this point clear.
* * *
**Your comment: Line 93: This is repetitive to the previous sentence.**

The introduction was re-written.
* * *
**Your comment: Section 1.3: Is all the description relevant for the paper? It is a bit long.**

The study area section was shortened.
* * *
**Your comment: Line 176 ff and in general: Did the authors consider the usage of any automated change detection approaches for image analysis? For some classes this might have been helpful and faster than digitizing all the features. For example, there are several studies and publications that successfully used such methods for post-earthquake damage assessment.**

We considered using automated method. Due to the limited quality of our data (e.g. Spot 5 image), large areas of bare surfaces created by landslides, vegetation that expanded above the houses, and dusts created by the earthquake in the 2008 image, simple classification methods could not extract the buildings. We do not have advanced commercial software for OOA analysis, thus we chose manual digitization. It is also the easiest way to ensure temporal consistency among the inventories, which would be hard for automatic approaches due to geo-referencing of multi-sourced images.
* * *
**Your comment: Figure 4: The color of the dormant landslides is not very well visible (it is better in the following figures). What is the dashed line in the image? There is only information later in the text, but not in the legend or the caption.**

In Figure 4 and 6, there were no dormant landslides because all of them were freshly triggered by the earthquake.

I removed dormant landslides from the legend in these two figures and added the thick dashed line.
* * *
**Your comment: Line 265: First sentence. What is the reason for that?**

It is caused by large numbers of the destroyed 1-floor WB buildings, which itself was more than all 2-floor buildings combined. This was a careless description.

This sentence is rewritten:

Overall the significance in damage ratio could only be observed in damage level 1. There were relatively more 1-floor buildings repairable (22%) than 2-floor buildings (11%). A difference related with

construction types was observed, as the survive rate of the RCM, WB, W types were 23%, 17%, and 9%. The damage ratios of the three major types (RCM, WB and W), are shown in Figure 5 A. There were only 4 RCF buildings and 2 survived.
* * *
*Technical corrections:*

**Your comment: In general, the paper is well written. However, spell check is needed, several typing errors need to be improved, partly formatting (chapter 1.3) should be adapted.**

Many sections are now rewritten. Section 1.3 was shortened
* * *
**Your comment: Table 4: The text "Sum: : :" is not readable.**

This was caused by an error when converting .docx file to .pdf file. I adjusted the line spacing of the table to make the contents visible.
* * *
**Comments of referee 2**

**Your comment: Moreover, if authors were to explain the results of their case study to someone in another country, what would they gain from this Chinese case study? Do they learn from the methodology applied?**

Reply: Other countries may learn the experiences and mistakes presented in this case than its methodology. A new section was added to discuss the cause of the problems and possible solutions.
* * *
**Your comment: In many phrases, authors provide facts, information, or ideas, but without supporting sources as to where those ideas or facts came from. All facts and information that are not common knowledge need to have citations (which are then put into the reference list) and all items in the reference list should be cited at least once in the**

**manuscript. Examples include but are not limited to: Section 1.1., lines 30-36 and 40, then further lines 270-284, and the events descriptions (e.g., in section 3.4) and also the final section 5.**

Reply:

References and sources (if obtained by interview and investigation) were added to these descriptions. In where literatures could not be found, the sentences were rephrased to express they were not certain or removed.
* * *
**Your comment: The niche and gap for this research have to be clearly addressed in section 1.1 so that the overall need and motivation for this work becomes clear, ideally this will go in lines 85-90 of the present manuscript.**

Reply:

a major revision of the introduction section was carried out. Sub-section 1.1 describes a general background. the descriptions about the Wenchuan earthquake went to a separate sub-section (1.2). The study area is now in the sub-section 1.3. The gap was described in the last paragraph of section 1.1.
* * *
**Your comment: Section 1.3 has to be shortened and included in section 1.2**

Reply:

We have revised this based on the comment of you and referee 1. The contents were shortened and merged
* * *
**Your comment: Figures: fonts are too small, Figures need north arrow, measured grid and maybe an inlet showing the case study area in China.**

Reply:

Fonts in legend was enlarged and all figures now have north arrow and grid. An inlet was added in figure 1
* * *
**Your comment: Figure 2: only Figure 2A is mentioned in the main text body.**

Reply: The description of construction types was accidentally removed in the manuscript. A new paragraph is added to describe construction types.
* * *
**Your comment: Please carefully check the English again, even if the material reads fluently, there are some minor errors such as debrisflow versus debris flow, etc.**

Reply: We corrected several spelling and grammar mistakes.
* * *
**Your comment: As already noted by referee #1, the overall text reads more like a technical report as a scientific paper. As such, I recommend to shorten the overall text by putting more information in Tables, and by combining some of the materials presented to that the overall appearance becomes more concise.**

Reply: Many of the result part was rewritten. We removed many unimportant text from the study area and reconstruction monitoring section. A discussion section was added to address the proposed gap and aim.